



# Vertical Dependence of Horizontal Variation of Cloud Microphysics: Observations from the ACE-ENA field campaign and implications for warm rain simulation in climate models

5    Zhibo Zhang[1,2,*], Qianqian Song[1,2], David B. Mechem[3], Vincent E. Larson[4], Jian Wang[5],

Yangang Liu[6], Mikael K. Witte[7,8], Xiquan Dong[9], Peng Wu[9,10]

1. Physics Department, University of Maryland Baltimore County (UMBC), Baltimore, 21250, USA

10   2. Joint Center for Earth Systems Technology, UMBC, Baltimore, 21250, USA

3. Department of Geography and Atmospheric Science, University of Kansas, Lawrence, 66045, USA

4. Department of Mathematical Sciences, University of Wisconsin — Milwaukee, Milwaukee, 53201, USA

15   5. Center for Aerosol Science and Engineering, Department of Energy, Environmental and Chemical Engineering, Washington University in St. Louis, St. Louis, 63130, USA

6. Environmental and Climate Science Department, Brookhaven National Laboratory, Upton, 11973, USA

7. Joint Institute for Regional Earth System Science and Engineering, University of
20   California Los Angeles, Los Angeles, 90095, USA

8. Jet Propulsion Laboratory, California Institute of Technology, Pasadena, 91011, USA

9. Department of Hydrology and Atmospheric Sciences, University of Arizona, Tucson, 85721, USA

10. Pacific Northwest National Laboratory, Richland, WA 99354, USA

To be submitted to the ACP special issue:  Marine aerosols, trace gases, and clouds over the North Atlantic

*Correspondence to*: Zhibo Zhang (zhibo.zhang@umbc.edu)





**Abstract:**

In the current global climate models (GCM), the nonlinearity effect of subgrid cloud variations on the parameterization of warm rain process, e.g., the autoconversion rate, is often treated by multiplying the resolved-scale warm ran process rates by a so-called enhancement factor (EF). In this study, we investigate the subgrid-scale horizontal variations and covariation of cloud water content ($q_c$) and cloud droplet number concentration ($N_c$) in marine boundary layer (MBL) clouds based on the in-situ measurements from a recent field campaign, and study the implications for the autoconversion rate EF in GCMs. Based on a few carefully selected cases from the field campaign, we found that in contrast to the enhancing effect of $q_c$ and $N_c$ variations that tends to make EF>1, the strong positive correlation between $q_c$ and $N_c$ results in a suppressing effect that makes tends to make EF<1. This effect is especially strong at cloud top where the $q_c$ and $N_c$ correlation can be as high as 0.95. We also found that the physically complete EF that accounts for the covariation of $q_c$ and $N_c$ has a robust decreasing trend from cloud base to cloud top. Because the autoconversion process is most important at the cloud top, this vertical dependence of EF should be taken into consideration in the GCM parametrization scheme.






## 1. Introduction

Marine boundary layer (MBL) clouds cover about 1/5 of Earth's surface and play an important
role the climate system *(Wood, 2012)*. A faithful simulation of MBL clouds in the global climate
model (GCM) is critical for the projection of future climate *(Bony and Dufresne, 2005; Bony et*
*al., 2015; Boucher et al., 2013)* and understanding of aerosol-cloud interactions *(Carslaw et al.,*
*2013; Lohmann and Feichter, 2005)*. Unfortunately, it turns out to be an extremely challenging
task. Among others, an important reason is that many physical processes in MBL clouds occur at
the spatial scales much smaller than the typical resolution of GCMs, making the simulation of
these processes in GCMs highly challenging.
Of particular interest in this study is the warm rain process that play an important role in
regulating the lifetime, water budget, and therefore integrated radiative effects of MBL clouds. In
the bulk cloud microphysics schemes that are widely used in GCMs *(Morrison and Gettelman,*
*2008)*, continuous cloud particle spectrum is often divided into two modes. Droplets smaller than
the "separation size" $r^*$ are classified into the cloud mode, which is described by two moments of
droplet size distribution (DSD), the droplet number concentration $N_c$ (0th moment of DSD) and
droplet liquid water content $q_c$ (proportional to the 3rd moment). Droplets larger than $r^*$ are
classified into a precipitation mode (drizzle or rain), with properties denoted by drop concentration
and water content ($N_r$ and $q_r$). In a bulk microphysics scheme, the transfer of mass from the cloud
to rain modes as a result of the collision-coalescence process is separated into two terms,
autoconversion and accretion: $\left(\frac{\partial q_r}{\partial t}\right)_{coal} = \left(\frac{\partial q_r}{\partial t}\right)_{auto} + \left(\frac{\partial q_r}{\partial t}\right)_{acc}$. Autoconversion is defined as the
rate of mass transfer from the cloud to rain mode due to the coalescence of two cloud droplets with
$r < r^*$. Accretion is defined as the rate of mass transfer due to the coalescence of a rain drop with



$r > r^*$ with a cloud droplet. A number of autoconversion and accretion parameterizations have
been developed, formulated either through numerical fitting of droplet spectra obtained from bin
microphysics LES or parcel model *(Khairoutdinov and Kogan, 2000)*, or through an analytical
simplification of the collection kernel to arrive at expressions that link autoconversion and
accretion with the bulk microphysical variables *(Liu and Daum, 2004)*. For example, a widely used
scheme developed by Khairoutdinov and Kogan (2000) ("KK scheme" hereafter) relates the
autoconversion with $N_c$ and  $q_c$ as follows:

$$\left(\frac{\partial q_r}{\partial t}\right)_{auto} = f_{auto}(q_c, N_c) = C q_c^{\beta_q} N_c^{\beta_N}, \qquad (1)$$

where $q_c$ and $N_c$ have units of kg kg–1 and cm–3, respectively; the parameter $C = 1350$, and the
two exponents $\beta_q = 2.47, \beta_N = -1.79$ are obtained through a nonlinear regression between the
variables $q_c$ and $N_c$ and the autoconversion rate derived from large-eddy simulation (LES) with
bin-microphysics spectra.

Having a highly accurate microphysical parameterization — specifically, highly accurate

local microphysical process rates — is not sufficient for an accurate simulation of warm-rain
processes in GCMs.  Clouds can have significant structures and variations at the spatial scale much
smaller than the typical grid size of GCMs (10 ~ 100 km) *(Barker et al., 1996; e.g., Cahalan and*
*Joseph, 1989; Lebsock et al., 2013; Wood and Hartmann, 2006; Zhang et al., 2019)*. Therefore,
GCMs need to account for these subgrid-scale variations in order to correctly calculate grid-mean
autoconversion and accretion rates. Pincus and Klein (2000) nicely illustrate this dilemma. Given
subgrid-scale variability represented as a distribution $P(x)$ of some variable $x$, for example the $q_c$
in Eq. (1), a grid-mean process rate is calculated as $\langle f(x) \rangle = \int f(x)P(x)dx$ , where $f(x)$ is the
formula for the local process rate. For nonlinear process rates such as autoconversion and accretion,
the grid-mean process rates calculated from the subgrid-scale variability does not equal the process





rate calculated from the grid-mean value of $x$, i.e., $\langle f(x) \rangle \neq f(\langle x \rangle)$. Therefore, calculating
autoconversion and accretion from grid-mean quantities introduces biases arising from subgrid-
scale variability. To take this effect into account, a parameter $E$ is often introduced as part of the
parameterization such that $\langle f(x) \rangle = E \cdot f(\langle x \rangle)$. Following the convention of previous studies, $E$ is
referred to as the "enhancement factor" (EF) here. Given the autoconversion parameterization
scheme, the magnitude of EF is primarily determined by cloud horizontal variability within a GCM
grid. Unfortunately, because most GCMs do not resolve subgrid cloud variation, the value of EF
is often simply assumed to be a constant for the lack of better options. The EF for KK
autoconversion scheme due to subgrid $q_c$ variation is assumed to be 3.2 in the two-moment scheme
by Morrison and Gettelman (2008), which is employed in the widely used Community Atmosphere
Model (CAM).

A number of studies have been carried out to better understand the horizontal variations of

cloud microphysics in MBL cloud and the implications for warm rain simulations in GCMs. Most
of these studies have been focused on the subgrid variation of $q_c$. Morrison and Gettelman (2008)
and several later studies *(Boutle et al., 2014; Hill et al., 2015; Lebsock et al., 2013; Zhang et al.,*
*2019)* showed that the subgrid variability $q_c$ and thereby the EF are dependent on cloud regime
and cloud fraction ($f_c$). They are generally smaller over the closed-cell stratocumulus regime with
higher $f_c$ and larger over the open-cell cumulus regime that often has a relatively small $f_c$. The
subgrid variance of $q_c$ is also dependent on the horizontal scale ($L$) of a GCM grid. Based on the
combination of in situ and satellite observations, Boutle et al. *(2014)* found that the subgrid $q_c$
variance first increases quickly with $L$ when $L$ is below about 20 km, then increases slow and
seems to approach to a asymptotic value for larger $L$. Similar spatial dependence is also reported
in Huang et al. *(2014),Huang and Liu (2014),* Xie and Zhang *(2015),* and Wu et al. *(2018)* which





are based on the ground radar retrievals from the Department of Energy (DOE) Atmospheric
Radiation Measurement (ARM) sites. The cloud-regime and horizontal-scale dependences have
inspired a few studies to parameterize the subgrid $q_c$ variance as a function of either $f_c$ or $L$ or a
combination of the two *(e.g., Ahlgrimm and Forbes, 2016; Boutle et al., 2014; Hill et al., 2015;*
*Xie and Zhang, 2015; Zhang et al., 2019).*

The aforementioned studies have an important limitation. They consider only the impacts

of subgrid  $q_c$ variations on the EF but ignore the impacts of subgrid variation of $N_c$ and its
covariation with $q_c$. Based on cloud fields from large-eddy simulation, Larson and Griffin *(2013)*
and later Kogan and Mechem *(2014; 2016)* elucidated that it is important to consider the
covariation of $q_c$ and $N_c$ to derive a physically complete and accurate EF for the autoconversion
parameterization. Lately, on the basis of MBL cloud observations from the Moderate Resolution
Imaging Spectroradiometer (MODIS) Zhang et al. *(2019)* (hereafter referred to as Z19) elucidate
that the subgrid variation of $N_c$ tends to further increase the EF for the autoconversion process in
addition to the EF due to $q_c$ variation. The effect of $q_c$-$N_c$ covariation on the other hand depends
on the sign of the $q_c$-$N_c$ correlation. A positive $q_c$-$N_c$ correlation would lead to an EF <1 that
partly offsets the effects of $q_c$ and $N_c$ variations. Although Z19 shed important new light on the
EF problem for the warm rain process, their study also suffers from limitations due to the use of
satellite remote sensing data. First, as a passive remote sensing technique, MODIS cloud product
can only retrieve the column-integrated cloud optical thickness and the cloud droplet effective
radius at cloud top, from which the column-integrated cloud liquid water path (LWP) is estimated.
As a result of using LWP, instead vertically resolved observations the vertical dependence of the
$q_c$ and $N_c$ horizontal variabilities are ignored in Z19. Second, the $N_c$ retrieval from MODIS is
based on several important assumptions, which can lead to large uncertainties (see review by



*(Grosvenor et al., 2018)* ). Furthermore, MODIS cloud retrieval product is known to suffer from
several inherent uncertainties, such as the three-dimensional radiative effects*(e.g., Zhang and*
*Platnick, 2011; Zhang et al., 2012; 2016)*, which in turn can lead to large uncertainties in the
estimated EF.
This study is a follow up of Z19. To overcome the limitations of satellite observations, we
use the in situ measurements of MBL cloud from a recent DOE field campaign, the Aerosol and
Cloud Experiments in the Eastern North Atlantic (ACE-ENA), to investigate the subgrid variations
of $q_c$ and $N_c$, as well as their covariation, and the implications for the simulation of autoconversion
simulation in GCMs. A main focus of this investigation is to understand the vertical dependence
of the $q_c$ and $N_c$ horizontal variations within the MBL clouds. This aspect has been neglected in
Z19 as well as most previous studies *(Boutle et al., 2014; Lebsock et al., 2013; Xie and Zhang,*
*n.d.)*. A variety of microphysical processes, such as adiabatic growth, collision-coalescence,
entrainment mixing, can influence the vertical structure of MBL clouds. At the same time, these
processes also vary horizontally at the subgrid scale of GCMs. As a result, the horizontal variations
of $q_c$ and $N_c$, as well as their covariation, and therefore the EFs may depend on the vertical location
inside the MBL clouds. It is important to understand this dependence for several reasons. First, the
warm rain process is usually initialized at cloud top where the autoconversion process of the cloud
droplets gives birth to embryo drizzle drops. The accretion process is, on the other hand, more
important in the lower part of the cloud *(Wood, 2005b)*. Thus, a better understanding of the vertical
dependence of horizontal variations of $q_c$ and $N_c$ inside of MBL cloud could help us understand
how the EF should be modeled in the GCMs. Second, a good understanding of the vertical
dependence of $q_c$ and $N_c$ variation inside of MBL clouds will also help us understand the
limitations in the previous studies, such as Z19, that use the column-integrated products for the





study of EF. Finally, this investigation may also be useful for modeling other processes, such as
aerosol-cloud interactions, in the GCMs.

Therefore, our main objectives in this study are to: 1) better understand the horizontal

variations of $q_c$ and $N_c$, as well as their covariation in MBL clouds, in particular their dependence
on the vertical height in cloud; 2) elucidate the implications for the EF of the autoconversion
parameterization in GCMs. The rest of the paper is organized as follows: we will describe the data
and observations used in this study in Section 2 and explain how we select the cases from the ACE-
ENA campaign for our study in Section 3. We will present cases studies in Section 4 and 5. Finally,
the results and findings from this study will be summarized and discussed in Section 6.

**2.  Data and Observations**

The data and observations used for this study are from two main sources: the in-situ

measurements from the ACE-ENA campaign and the ground-based observations from the ARM
ENA site. The ENA region is characterized by persistent subtropical MBL clouds that are
influenced by different seasonal meteorological conditions and a variety of aerosol sources *(Wood*
*et al., 2015)*. A modeling study by Carslaw et al. *(2013)* found the ENA to be one of regions over
the globe with the largest uncertainty of aerosol indirect effect. As such, the ENA region attracted
substantial attention over the past few decades for aerosol-cloud interaction studies. From April
2009 to December 2010 the DOE ARM program deployed its ARM Mobile Facility (AMF) to the
Graciosa Island (39.09°N, 28.03°W) for a measurement field campaign targeting the properties of
cloud, aerosol and precipitation in the MBL (CAP-MBL) in the Azores region of ENA *(Wood et*
*al., 2015)*. The measurements from the CAP-MBL campaign have proved highly useful for a
variety of purposes, from understanding the seasonable variability of clouds and aerosols in the





MBL of the ENA region *(Dong et al., 2014; Rémillard et al., 2012)* to improving cloud
parameterizations in the GCMs *(Zheng et al., 2016)* and to validating the space-borne remote
sensing products of MBL clouds *(Zhang et al., 2017)*. The success of the CAP-MBL revealed that
the ENA has an ideal mix of conditions to study the interactions of aerosols and MBL clouds. In
2013 a permanent measurement site is established by the ARM program on Graciosa Island as its
newest permanent atmospheric observatories, also known as the ENA site *(Voyles and Mather,*
*2013)*.
**2.1. In situ measurements from the ACE-ENA campaign**

The Aerosol and Cloud Experiments in ENA (ACE-ENA) project was "*motivated by the*

*need for comprehensive in situ characterizations of boundary-layer structure and associated*
*vertical distributions and horizontal variabilities of low clouds and aerosol over the Azores*"
*(Wang et al., 2016)*. The ARM Aerial Facility (AAF) Gulfstream-1 (G-1) aircraft was deployed
during two intensive measurement periods (IOPs), the summer 2017 IOP from June 21 to July 20,
2017 and the winter 2018 IOP from January 15 to February 18, 2018. Over 30 research flights (RF)
were carried out during the two IOPs around the ARM ENA site on Graciosa Island that sampled
a large variety of cloud and aerosol properties along with the meteorological conditions.

Table 1 summarizes the in-situ measurements from the ACE-ENA campaign used in this

study. The location and velocity of G1 aircraft, and the environment meteorological conditions
during the flight (temperature, humidity, and wind velocity) are taken from Aircraft-Integrated
Meteorological Measurement System 20-Hz (AIMMS-20) dataset *(Beswick et al., 2008)*. The size
distribution of cloud droplets, and the corresponding $q_c$ and $N_c$ are obtained from the fast cloud
droplet probe (FCDP) measurement. The FCDP measures the concentration and size of cloud
droplets in the diameter size range from 1.5 to 50 µm in 20 size bins with an overall uncertainty





of size around 3 µm *(Lance et al., 2010; SPEC, 2019)*. Following previous studies *(Wood, 2005a)*,
we adopt a $r^* = 20 \ \mu m$ as the threshold to separate cloud droplets from drizzle drops, i.e., drops
with $r < r^*$ are considered as cloud droplets. After the separation, the $q_c$ and $N_c$ are derived from
the FCDP droplet size distribution measurements. As an evaluation, we compared our FCDP-
derived $q_c$ results with the direct measurements of $q_c$ from the multi-element water content system
(WCM-2000) also flown during the ACE-ENA and found an excellent agreement. We also
performed a couple of sensitivity tests in which we perturbed the value of $r^*$ by $\pm 5$ µm. The
perturbation shows little impact on the results shown in sections 4 and 5. The cloud droplet
spectrum from the FCDP is available at a frequency of 10 Hz. Since the typical horizontal speed
of the G-1 aircraft during the in-cloud leg is about 100 m s-1, the spatial sampling rate these
instruments is on the order of 10 m for the FCDP.
**2.2. Ground observations from ARM ENA site**
In addition to the in-situ measurements, ground measurements from the ARM ENA site
are also used to provide ancillary data for our studies. In particular, we will use the Active Remote
Sensing of Cloud Layers product (ARSCL; *(Clothiaux et al., 2000; Kollias et al., 2005)* which
blends radar observations from the Ka–band ARM zenith cloud radar (KAZR), micropulse lidar
(MPL), and the ceilometer to provide information on cloud boundaries and the mesoscale structure
of cloud and precipitation. The ARSCL product is used to specify the vertical location of the G1
aircraft and thereby the in-situ measurements with respect to the cloud boundaries, i.e., cloud base
and top (see example in Figure 1). In addition, the radar reflectivity observations from KAZR,
alone with in situ measurements, are used to select the precipitating cases for our study. Note that
the ARSCL product is from the vertically pointing instruments, which sometimes are not





collocated with the in-situ measurements from G1 aircraft. As explained later in the next section,
only those cases with a reasonable collocation are selected for our study.

**3. Case selections**

**3.1. ACE-ENA flight pattern**

The section provides a brief overview of the G1 aircraft flight patterns during the ACE-
ENA and explains the method for cases selections for our study using the July 18, 2017 RF as an
example. As shown in Table 2, a variety of MBL conditions were sampled during the two IOPs of
the ACE-ENA campaign, from mostly clear-sky to thin stratus and drizzling stratocumulus. In this
study, we are interested in the RFs that encountered the drizzling stratocumulus clouds, since our
objective is to understand the implications of subgrid cloud variation for the autoconversion
process. The basic flight patterns of G1 aircraft in the ACE-ENA included spirals to obtain vertical
profiles of aerosol and clouds, and legs at multiple altitudes, including below cloud, inside cloud,
at the cloud top, and in the free troposphere. As an example, Figure 1a shows the horizontal
location of the G1 aircraft during the July 18, 2017 RF which is the "golden case" for our study as
explained in the next section. The corresponding vertical track of the aircraft is shown in Figure
1b overlaid on the reflectivity curtain of ground based KAZR. In this RF, the G1 aircraft repeated
multiple times of horizontal level runs in a "V" shape at different vertical levels inside, above and
below the MBL (see Figure 1b). The lower tip of the "V" shape is located at the ENA site on
Graciosa island. The average wind in the upper MBL (i.e., 900 mb) is approximately Northwest.
So, the left side of the V-shape horizontal level runs is along the wind and the right side cross the
wind. Note that the horizontal velocity of the G1 aircraft is approximately 100 m s$_{-1}$. Since the
duration of these selected "V" shape hlegs is between 580 s and 700 s, their total horizontal length
is roughly 60 km, with each side of the "V" shape ~30 km. These "V" shape horizontal level runs,



with one side along and the other cross the wind, are a common sampling strategy used in the
ACE-ENA to observe the properties of aerosol and cloud at different vertical levels of the MBL.
In our study we use the vertical location of the G1 aircraft from the AIMMS to identify continuous
horizontal flight tracks which are referred to as the "hleg". For the July 18, 2017 case, a total of 13
hlegs are identified as shown in Figure 1b. Among them, the hleg 5, 6, 7, 8, 10, 11, and 12 are the
seven V-shape horizontal level runs inside the MBL cloud. Together they provide an excellent set
of samples of the MBL cloud properties at different vertical levels of a GCM grid box of about 30
km. As aforementioned, Boutle et al. (2014) found that the horizontal variance of $q_c$ increases with
the horizontal scale $L$ slowly when $L$ is larger than about 20 km. Therefore, although the horizontal
sampling of the selected hlegs is only about 30 km, the lessons learned here could yield useful
insights for larger GCM grid sizes. In addition to the hlegs, we also identified the vertical
penetration legs in each flight, referred to as the "vlegs", from which we will obtain the vertical
structure of the MBL, along with the properties of cloud and aerosol.
**3.2. Case selection**
As illustrated in Figure 1 a and b for the July 18, 2017 RF, the criterions we used to select
the RF cases and the hlegs within the RF can be summarized as follows:
• The RF encounters precipitating MBL clouds according to both pilot report and radar
reflectivity observations from the ground-based KAZR.
• The RF samples multiple continuous in-cloud hlegs at different vertical levels with the
horizontal length of at least 10 km and cloud fraction larger than 10% (i.e., the fraction of
a hleg with $q_c$>0.01g m$_{-3}$ must exceed 10% of the total length of that hleg)
• Moreover, the selected hlegs must sample the same region repeatedly in terms of horizontal
track but different vertical levels in terms of vertical track. Take the July 18, 2017 case as





an example. The hleg 5, 6, 7, 8 follow the same "V" shape horizontal track (see Figure 1a)
but sample different vertical levels of the MBL clouds (see Figure 1b). Such hlegs provide
us the horizontal sampling needed to study the subgrid horizontal variations of the cloud
properties and, at the same time, the chance to study the vertical dependence of the
horizontal cloud variations.
• Finally, the RF needs to have at least one vleg and the cloud boundary derived from the
vleg is largely consistent with that derived from the ground-based measurements. This
requirement is to ensure that the vertical locations of the selected hlegs with respect to
cloud boundaries can be specified. For example, as shown in Figure 1b according to the
ground-based observations, the hlegs 5 and 10 of July 18, 2017 case are close to cloud base,
while hlegs 8 and 12 close to cloud top (see also Figure 4).
The above requirements together pose a strong constraint on the observation. Fortunately, thanks
to the careful planning of the RF which had already taken studies like ours into consideration, we
are able to select a total of four RF cases as summarized in Table 3. We will first focus on the
"golden case—July 18, 2017 RF and then investigate if the lessons learned from the July 18, 2017
RF also apply to the other three cases.
**4. A study of the July 18, 2017 case**
**4.1. Horizontal and vertical variations of cloud microphysics**
On July 18, 2017, the North Atlantic is controlled by the Icelandic low to the north and the
Azores high to the south (see Figure 2b), which is a common pattern of large-scale circulation
during the summer season in this region *(Wood et al., 2015)*. The Azores is at the southern tip of
the cold air sector of a frontal system where the fair-weather low-level stratocumulus clouds are
dominant (see satellite image in Figure 2a). The RF on this day started around 8:30 UTC and ended



around 12 UTC. As explained in the previous section, we selected 7 hlegs from this RF that
horizontally sampled the same region repeatedly in a similar "V" shaped track but vertically at
different levels. The radar reflectivity observation from the ground based KAZR during the same
period peaks around 10 dBZ indicating the presence of significant drizzle inside the MBL clouds.

303     Among the 7 selected hlegs, the hlegs 5, 6, 7 and 8 are 4 consecutive "V" shape tracks,

with hlegs 5 close to cloud base and hleg 8 close to cloud top. The hlegs 10,11 and 12 are another
set of consecutive "V" shape tracks with hlegs 10 and 12 close to cloud base and top, respectively
(see Figure 1). Using $q_c > 0.01$ gm–3 as a threshold for cloud, the cloud fraction ($f_c$) of all these
hlegs is close to unity (i.e., overcast), except for the two hlegs close to cloud base ($f_c$=46% for
hleg 5 and $f_c$=51% for hleg 10). The $q_c$ and $N_c$ derived from the in situ FCDP measurements for
these selected hleg are plotted in Figure 3 as a function of UTC time. It is evident from Figure 3
that both $q_c$ and $N_c$ have significant horizontal variations. At cloud base (see Figure 3d for hleg 5
and Figure 3g for hleg 10) the $q_c$ varies from 0.01 gm–3 (i.e., the lower threshold) up to about 0.4
gm–3 and the $N_c$ from 25 cm–3 up to 150 cm–3, with the mean values around 0.08 gm–3 and 65
cm–3 , respectively. Such strong variations of cloud microphysics could be contributed by a number
of factors. One can see from the ground radar and lidar observations in Figure 1b that the height
of cloud base varies significantly. As a result, the horizonal legs may not really sample the cloud
base. In addition, the variability in updraft at cloud base cloud lead to the variability in the
activation and growth of cloud condensation nuclei (CCN). In the middle of the MBL cloud, i.e.,
hleg 6 (Figure 3c), 7 (Figure 3b) and 11 (Figure 3f), the mean value of $q_c$ is significantly larger
than that of cloud base hlegs while the variability is reduced. The mean value of $q_c$ keeps
increasing toward cloud top to ~0.73 gm–3 in hleg 8 (Figure 3a) and to ~0.53 gm–3 in hleg 12





(Figure 3e), respectively. In contrast, the horizontal variability of $q_c$ seems to increase in
comparison with those observed in mid-level hlegs.

To obtain a further understanding of the vertical variations of cloud microphysics, we

analyzed the cloud microphysics observations from the two green-shaded vlegs 1 and 3 in Figure
1b. The vertical profile of the mean $q_c$ and $N_c$ from these two vlegs are shown in Figure 4a and
Figure 4b, respectively, with over-plotted the mean and standard deviation of the $q_c$ and $N_c$
derived from the 7 selected hlegs. Overall, the vertical profiles of the $q_c$ and $N_c$ are qualitatively
aligned with the classic adiabatic MBL cloud structure *(Brenguier et al., 2000; Martin et al., 1994)*.
That is, the $N_C$ remains relatively a constant (see Figure 4b) while the $q_c$ increases approximately
linearly with height from cloud base upward as a result of condensation growth (see Figure 4a,),
except for the very top of the cloud, i.e., the entrainment zone where the dry air entrained from the
above mixes with the humid cloudy air in the MBL. In previous studies, a so-called inverse relative
variance, $\nu$, is often used to quantify the subgrid variations of cloud microphysics. It is defined as
follows

$$\nu_X = \frac{\langle X \rangle^2}{\sigma_X^2}, \tag{2}$$

where $X$ is either $q_c$ (i.e., $\nu_X = \nu_{q_c}$) or $N_c$ (i.e., $\nu_X = \nu_{N_c}$). $\langle X \rangle$ and $\sigma_X$ are the mean value and
standard deviation of $X$, respectively. As such the smaller the $\nu$ value the larger the horizontal
variation of $X$ in comparison with the mean value. As shown in Figure 4c, the $\nu_{q_c}$ and $\nu_{N_c}$ derived
from the selected hlegs follow a similar vertical pattern: they both increase first from cloud base
upward and then decrease in the entrainment zone, with the turning point somewhere around 1 km
(i.e., around hleg 7 and 11). It indicates that both $q_c$ and $N_c$ have significant horizontal variabilities
at cloud base which may be a combined result of horizontal fluctuations of dynamics (e.g., updraft)
and thermodynamics (e.g., temperature and dynamics), as well as horizontal variations of aerosols.



The horizontal variabilities of both $q_c$ and $N_c$ both decrease upward toward cloud top until the
entrainment zone where both variabilities increase again.
So far, in all the analyses above the variations of $q_c$ and $N_c$ have been considered
separately and independently. As pointed out in several previous studies, the co-variation of $q_c$
and $N_c$ could have an important impact on the EF for the autoconversion process in GCMs *(Kogan*
*and Mechem, 2016; Larson and Griffin, 2013; Zhang et al., 2019)*. This point will be further
elucidated in detail in the next section. Figure 5 shows the joint distributions of $q_c$ and $N_c$ for the
7 selected hlegs and the corresponding linear correlation coefficients as a function of height are
shown in Figure 4d. For the sake of reference, the linear correlation coefficient between $\ln(q_c)$
and $\ln(N_c)$, i.e., the $\rho_L$ that will be introduced later in Eq. *(4)*, is also plotted in Figure 4d.
Looking first at the hlegs 10, 11 and 12, i.e., the 2nd group of consecutive "V" shape legs, there is
a clear increasing trend of the correlation between $q_c$ and $N_c$ from cloud bottom ($\rho = 0.75$ for
hleg 10) to cloud top ($\rho = 0.95$ for hleg 12). The picture based on the hlegs 5, 6, 7, and 8 is more
complex. As shown in Figure 5, the joint distributions of $q_c$ and $N_c$ of hleg 6 (Figure 5b), hleg 7
(Figure 5c) and, to a less extent, hleg 8 (Figure 5d) all exhibit a clear bimodality. Further analysis
reveals that each of the two modes in these bimodal distributions approximately corresponds to
one side of the "V" shape track. As aforementioned, for all the selected 7 "V" shape hlegs, the left
side is along the wind and the right side across the wind (see Figure 1). To illustrate this difference,
the across-wind side of the hleg is shaded in yellow in Figure 3. It is intriguing to note that the $N_c$
from the across-wind side of the hleg are systematically larger than those from the along-wind side,
while their $q_c$ values are largely similar. As a result of this bimodality of $N_c$, the correlation
coefficients between $q_c$ and $N_c$ is significantly smaller for the hlegs 6 ($\rho = 0.22$) and 7 ($\rho = 0.31$)
in comparison with other hlegs. However, if the two sides of the "V" shape tracks are considered





separately, then the $q_c$ and $N_c$ become more correlated, except for the cross-wind side of hleg 6
which still exhibits to some degree a bimodal joint distributions of $q_c$ and $N_c$. In spite of the
bimodality, there is evidently a general increasing trend of the correlation between $q_c$ and $N_c$ from
cloud base toward cloud top. At the cloud top, the $q_c$ and $N_c$ correlation coefficient can be as high
as $\rho = 0.95$ for hleg 12 (see Figure 5e). As explain in the next section, this close correlation
between $q_c$ and $N_c$ has important implications for the simulation of autoconversion enhancement
factor.

As a summary, the above phenomenological analysis of the July 18, 2017 RF reveals the

following features of the horizontal and vertical variations of cloud microphysics. Vertically, the
mean values of $q_c$ and $N_c$ qualitatively follow the adiabatic structure of MBL cloud, i.e., $q_c$
increases linear with height and $N_c$ remains largely invariant above cloud base. Even though the
joint distribution of $q_c$ and $N_c$ exhibits a bimodality in several hlegs, their correlation generally
increases with height and can be as high as $\rho = 0.95$ at cloud top. Horizontally, both $q_c$ and $N_c$
have a significant variability at cloud base, which tends to first decrease upward and then increase
in the uppermost part of cloud close to the entrainment zone.
**4.2. Implications for the EF for the autoconversion rate parameterization**

As explained in the introduction, in GCMs the autoconversion process is usually

parameterized as a highly nonlinear function of $q_c$ and $N_c$, e.g., the KK scheme in Eq. (1). In such
parameterization, an EF is needed to account for the bias caused by the nonlinearity effect. A
variety of methods have been proposed and used in the previous studies to estimate the EF *(Larson*
*and Griffin, 2013; Lebsock et al., 2013; Pincus and Klein, 2000; Zhang et al., 2019)*. The methods
used in this study are based on Z19. Only the most relevant aspects are recapped here. Readers are
referred to Z19 for detail.





If the subgrid variations of $q_c$ and $N_c$, as well as their covariation, are known, then the EF
can be estimated based on its definition as follows

$$E = \frac{\int_0^\infty \int_0^\infty q_c^{\beta_q} N_c^{\beta_N} P(q_c, N_c) dq_c dN_c}{\langle q_c \rangle^{\beta_q} \langle N_c \rangle^{\beta_N}}, \qquad (3)$$

where $\langle q_c \rangle$ and $\langle N_c \rangle$ are the grid-mean value, $P(q_c, N_c)$ is the joint probability density function
(PDF) of $q_c$ and $N_c$. Some previous studies approximate the $P(q_c, N_c)$ as a bivariate lognormal
distribution as follows:

$$P(q_c, N_c) = \frac{1}{2\pi q_c N_c \sigma_{q_c} \sigma_{N_c} \sqrt{1 - \rho_L^2}} exp\left(-\frac{\zeta}{2}\right)$$

$$\zeta = \frac{1}{1 - \rho_L^2}\left[\left(\frac{lnq_c - \mu_{q_c}}{\sigma_{q_c}}\right)^2 - 2\rho\left(\frac{lnq_c - \mu_{q_c}}{\sigma_{q_c}}\right)\left(\frac{lnN_c - \mu_{N_c}}{\sigma_{N_c}}\right) + \left(\frac{lnN_c - \mu_{N_c}}{\sigma_{N_c}}\right)^2\right], \qquad (4)$$

where $\mu_X$ and $\sigma_X$ are, respectively, the mean and standard deviation of $\ln(X)$, where $X$ is either
$q_c$ or $N_c$. $\rho_L$ is the linear correlation coefficient between $\ln(q_c)$ and $\ln(N_c)$, *(Larson and Griffin,*
*2013; Lebsock et al., 2013; Zhang et al., 2019)*. It should be noted here that $\rho_L$ is fundamentally
different from $\rho$ (i.e., the linear correlation coefficient between $q_c$ and $N_c$). On the other hand, we
found that for all the selected hlegs $\rho$ and $\rho_L$ are in an excellent agreement (see Figure 4d). In fact,
$\rho$ and $\rho_L$ can be used interchangeably in the context of this study without any impact on the
conclusions. Nevertheless, interested readers may find more detailed discussion of the relationship
between $\rho$ and $\rho_L$ in Larson and Griffin *(2013)*.
Substituting $P(q_c, N_c)$ in Eq. (4) into Eq. (3) yields a formula for EF that consists of the
following three terms

$$E = E_q\left(\nu_{q_c}, \beta_q\right) \cdot E_N\left(\nu_{N_c}, \beta_N\right) \cdot E_{COV}\left(\rho_L, \beta_q, \beta_N \nu_{q_c}, \nu_{N_c}\right), \qquad (5)$$





where $E_q(v_{q_c}, \beta_q)$ corresponds to the enhancing effect of the subgrid variation of $q_c$, if $q_c$ follows
a marginal lognormal distribution, i.e., $P(x) = \frac{1}{\sqrt{2\pi}x\sigma} \exp\left(-\frac{(\ln x - \mu)^2}{2\sigma^2}\right)$. It is a function of the
inverse relative variance $v_q$ in Eq. (2) as follows:

$$E_q(v_{q_c}, \beta_q) = \left(1 + \frac{1}{v_{q_c}}\right)^{\frac{\beta_q^2 - \beta_q}{2}}. \tag{6}$$

Similarly, the $E_N(v_{N_c}, \beta_N)$ below corresponds to the enhancing effect of the subgrid variation of
$N_c$, if $N_c$ follows a marginal lognormal distribution,

$$E_N(v_{N_c}, \beta_N) = \left(1 + \frac{1}{v_{N_c}}\right)^{\frac{\beta_N^2 - \beta_N}{2}}. \tag{7}$$

The third term $E_{COV}(\rho_L, \beta_q, \beta_N v_{q_c}, v_{N_c})$ in Eq. (5)

$$E_{COV}(\rho_L, \beta_q, \beta_N, v_{q_c}, v_{N_c}) = exp(\rho_L \beta_q \beta_N \sigma_{q_c} \sigma_{N_c}), \tag{8}$$

corresponds to the impact of the co-variation of $q_c$ and $N_c$ on the EF. Because $\beta_q > 0$ and $\beta_N <$
0, if $q_c$ and $N_c$ are negatively correlated (i.e., $\rho_L < 0$) then the $E_{COV} > 1$ and acts as an enhancing
effect on the autoconversion rate computation. In contrast, if $q_c$ and $N_c$ are positively correlated
(i.e., $\rho_L > 0$), then the $E_{COV} < 1$ which becomes a suppressing effect on the autoconversion rate
computation.
As aforementioned, most previous studies of the EF consider only the impact of subgrid
$q_c$ variation (i.e., only the $E_q$ term). The impacts of subgrid $N_c$ variation as well as its covariation
with $q_c$ have been largely overlooked in observational studies, in which, the $E_q$ is often derived
from the observed subgrid variation of $q_c$ based on the definition of EF, i.e.,



$$E_q = \frac{\int_0^\infty q_c^{\beta_q} P(q_c) dq_c}{\langle q_c \rangle^{\beta_q}}, \tag{9}$$

where $P(q_c)$ is the observed subgrid PDF of $q_c$. Alternatively, $E_q$ have also been estimated from
the inverse relative variance $\nu_q$ by assuming the subgrid variation of $q_c$ to follow either the
lognormal distribution, in which case $E_q$ is given in Eq. (6).

Similar to $E_N$, if only the effect of subgrid $N_c$ is considered, the corresponding $E_N$ can be

derived from the following two ways, one from the observed subgrid PDF $P(N_c)$ based on the
definition of EF, i.e.,

$$E_N = \frac{\int_0^\infty N_c^{\beta_N} P(N_c) dN_c}{\langle N_c \rangle^{\beta_N}}, \tag{10}$$

and the other based on Eq. (7) from the relative variance $\nu_{N_c}$ by assuming the subgrid $N_c$
variation to follow the lognormal distribution.

Now, we put the in-situ $q_c$ and $N_c$ observations from the selected hlegs in the theoretical

framework of EF described above and investigate the following questions:

1) What is the ("observation-based") EF derived based on Eq. (3) from the observed joint

PDF $P(q_c, N_c)$?

2) How well does the ("bi-logarithmic") EF derived based on Eq. (5) by assuming that

the covariation of $q_c$ and $N_c$ follows a bi-variate lognormal agree with the observation-based EF?

3) What is the relative importance of the $E_q$, $E_N$, and $E_{COV}$ terms in Eq. (5) in

determining the value of EF?

4) What is the error of considering only $E_q$ and omitting the $E_N$ and $E_{COV}$ terms?

5) How do the observation-based EFs from Eq. (3) and the $E_q$, $E_N$, $E_{COV}$ terms vary with

vertical height in cloud?


These questions are addressed in the rest of this section. Focusing first on the $E_q$ in Figure
6a, the $E_q$ derived from observation based on Eq. (9) (solid circle) shows a clear decreasing trend
with height between cloud base at around 700 m to about 1 km, with value reduced from about 3
to about 1.2. Then, the value of $E_q$ increases slightly in the cloud top hlegs 8 and 12. The $E_q$
derived based on Eq. (6) by assuming lognormal distribution (open circle) has a very similar
vertical pattern, although the value is slightly overestimated in comparison with the observation-
based result. The vertical pattern of $E_q$ can be readily explained by how the subgrid variation of
$q_c$ in Figure 4c. The $E_N$ derived from observation (solid triangle) in Figure 6b shows a similar
vertical pattern as $E_q$, i.e., first decreasing with height from cloud base to about 1.2 km and then
increasing with height in the uppermost part of cloud. The $E_N$ derived based on Eq. (7) by
assuming a lognormal distribution (open triangle) show a large error compared with the
observation-based values, especially at cloud base (i.e., hleg 5 and 10) and cloud top (i.e., hleg 8
and 12).
Using hleg10 as an example, we further investigated the cause for the error in lognormal-
based EFs in comparison with those diagnosed from the observation. As shown in Figure 7a the
observed $q_c$ is slightly negatively skewed in logarithmic space by the small values. Because the
autoconversion rate is proportional to $q_c^{2.47}$, the negatively skewed $q_c$ also leads to a negatively
skewed $E_q$ in Figure 7b. As a result, the leg-averaged $E_q$ diagnosed from the observation is slightly
smaller than that derived based on Eq. (6) by assuming a lognormal distribution. The negative
skewness also explains the large error in $E_N$ for hleg 10 seen on Figure 6b. As shown in Figure 7c
the observed $N_c$ is also negatively skewed, to a much larger extent in comparison with $q_c$. Because
the autoconversion rate is proportional to $N_c^{-1.79}$, the highly negatively skewed $N_c$ results in a





highly *positively* skewed $E_N$ in Figure 7d. As a result, the $E_N$ diagnosed from the observation is
much larger than that derived based on Eq. (7) by assuming a lognormal distribution.

The $E_q$ and $E_N$ reflect only the individual contributions of subgrid $q_c$ and $N_c$ variations to

the EF. The effect of the covariation of $q_c$ and $N_c$, i.e., the $E_{COV}$ is shown in Figure 6c.
Interestingly, the value of $E_{COV}$ is smaller than unity for all the selected hlegs. As explained in Eq.
(8), $E_{COV} < 1$ is a result of a positive correlation between $q_c$ and $N_c$, as seen in Figure 4d.
Therefore, in these hlegs the covariation of the $q_c$ and $N_c$ has *suppressing* effect on the EF, in
contrast to the enhancing effect of $E_q$ and $E_N$. This result is qualitatively consistent with Z19 who
found that the vertically integrated liquid water path (LWP) of MBL clouds is in general positively
correlated with the $N_c$ estimated from the MODIS cloud retrieval product and, as a result, $E_{COV} <$
1 over most of the tropical oceans. Because of the relationship in Eq. (8), the value $E_{COV}$ is
evidently negatively proportional to the correlation coefficient $\rho_L$ in Figure 4d. The largest value
is seen in hleg 6 and 7 in which the bimodal joint distribution of $q_c$ and $N_c$ results in a small $\rho_L$.
The smallest $E_{COV} = 0.21$ is seen in hleg 12, as result of a strong correlation between $q_c$ and $N_c$
($\rho_L = 0.96$) and moderate $\sigma_q$ and $\sigma_N$.

Finally, the EF that accounts for all factors, including the individual variations of $q_c$ and

$N_c$, as well as their covariation, is shown in Figure 6d. Focusing first on the observation-based
results (solid star), i.e., $E$ in Eq. (3), evidently there is a decreasing trend from cloud base (e.g.,
$E = 2.2$ for hleg 5 and $E = 1.59$ for hleg 10) to cloud top (e.g., $E = 1.20$ for hleg 8 and $E = 1.02$
for hleg 12). The $E$ derived based on Eq. (5) by assuming the bi-variate lognormal distribution
between $q_c$ and $N_c$ (i.e., open star in Figure 6d) are generally larger than the observation-based
results, in particularly for hleg 6 and 7. To investigate the reason for this error, we compared the
observed joint PDF between $q_c$ and $N_c$ for hleg 7 with the diagnosed bi-variate lognormal



distribution in Figure 8. As already noted, the observed $q_c$ and $N_c$ follow a bimodal distribution
which leads to a rather small correlation coefficient $\rho_L$. The bi-variate lognormal distribution
interprets this small $\rho_L$ as an abroad unimodal distribution (dashed contour line), which leads to
an overestimate of EF.

Finally, it is intriguing to note that the value of $E = E_q \cdot E_N \cdot E_{COV}$ in Figure 6d is

comparable to $E_q$ Figure 6a, which indicates that the enhancing effect of $E_N > 1$ in Figure 6b is
partially canceled by the suppressing effect of $E_{COV} < 1$ in Figure 6c. As aforementioned, many
previous studies of the EF consider only the effect of $E_q$ but overlook the effect of $E_N$ and $E_{COV}$.
The error in the studies would be quite large if it were not for a fortunate error cancellation.
**5.  Other Selected Cases**

In addition to the July 18, 2017 RF, we also found another 3 RFs that meet our criterions

as described in Section 3 for case selection. As summarized in Table 3 and shown in Figure 1c-h,
the July 20, 2017 and Jan. 19, 2018 RFs sampled the MBL clouds around the ENA site repeatedly
in a "V" shape horizontal pattern similar to the July 18, 2017 RF. In contrast, the Feb. 11, 2018
RF is different from the other three cases in two aspects. First, its horizontal sampling pattern is a
simple straight line. Second, the boundary layer is significantly deeper, with a mean cloud top
height around 1.5 km in comparison to the ~ 1 km cloud top height in other RFs. Due to limited
space, we cannot present the detailed case studies of these RFs. Instead, we view them collectively
and investigate whether the lessons learned from the July 18, 2017 RF, especially those about the
EF in Section 4.2, also apply to the other cases.

In order to compare the hlegs from different RFs, we first normalize the altitude of each

hleg with respect to the minimum and maximum values of all selected hlegs in each RF as
follows:





$$z^*_{hleg} = \frac{z_{hleg} - z_{min}}{z_{max} - z_{min}}, \qquad\qquad (11)$$

where $z^*_{hleg}$ is the normalized altitude for each hleg in a RF, $z_{min}$ and $z_{max}$ are the altitude of the
lowest and highest hleg in the corresponding RF. Defined this way, $z^*_{hleg}$ is bounded between 0
and 1. Alternatively, $z^*_{hleg}$ could also be defined with respect to the averaged cloud top ($z_{top}$) and
base ($z_{base}$) as inferred from the KAZR or vlegs. However, because of the variation of cloud top
and cloud base heights, as well as the collocation error, the $z^*_{hleg}$ would often become significantly
larger than 1 or smaller than 0, if $z^*_{hleg}$ were defined with respect to $z_{top}$ and $z_{base}$, making results
confusing and difficult to interpret.

Figure 9 shows the observation based EFs for all the selected hlegs from the 4 selected RFs

as a function of the $z^*_{hleg}$. As shown in Figure 9a, the $E$ derived based on (*3*) that accounts for the
covariation of $q_c$ and $N_c$ has a decreasing trend from cloud base to cloud top. This is consistent
with the result from the July 18, 2017 case in Figure 6d. However, neither the $E_q$ in Figure 9b nor
the $E_N$ in Figure 9c shows a clear dependence on $z^*_{hleg}$ in comparison with the results of July 18th,
2017 case in Figure 6a and b. Note that the $E_q$ and $E_N$ are influenced by a number of factors, such
as horizontal distance and cloud fraction, in addition to vertical height. It is possible that the
differences in other factors outweigh the vertical dependence here. Interestingly, the linear
correlation coefficient $\rho$ between $q_c$ and $N_c$ in Figure 9d shows an increasing trend with $z^*_{hleg}$ that
is statistically significant (R-value= 0.50 and P-value=0.02), despite a few outliers including the
aforementioned hleg 6 an 7 from July 18, 2017 case and also the hleg 16 from Jan. 19, 2018 case.
It turns out that the joint distribution of $q_c$ and $N_c$ in the hleg 16 of the Jan. 19, 2018 is also bimodal
(similar to Figure 5b and not shown here), leading to a small $\rho_L$. Nevertheless, the increasing trend
of $\rho$ with $z^*_{hleg}$ in Figure 9d is consistent with what we found in the July 18, 2017 case (see Figure

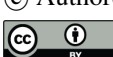



4d). As evident from Eq. ($8$), an increase of $\rho_L$ would lead to a decrease of $E_{COV}$. Since neither $E_q$
nor $E_N$ shows a clear dependence on $z^*_{hleg}$, the decrease of $E_{COV}$ with $z^*_{hleg}$ seems to play an
important role in the determining the value of $E$. Another line of evidence supporting this role is
the fact that both $E_q$ and $E_N$ are quite large for the cloud top hlegs, while in contrast the values of
corresponding $E$ that accounts for the covariation of $q_c$ and $N_c$ are much smaller. For example, the
$E_q$ for two hlegs from the Feb. 11, 2018 RF exceeds 8 but the corresponding $E$ values are smaller
than 1.2 which is evidently a result of large $\rho_L$ and thereby small $E_{COV}$.
As aforementioned, many previous studies of the EF for the autoconversion rate
parameterization consider only the effect of subgrid $q_c$ variation but ignore the effects of subgrid
$N_c$ variation, and its covariation with $q_c$. To understand the potential error, we compared the $E_q$
and $E$ both derived based on observations in Figure 10. Apparently, $E_q$ is significantly larger than
$E_q$ for most of the selected hlegs, which implies that the considering only subgrid $q_c$ variation
would likely lead to an overestimation of EF. This is an interesting result. Note that $E_N \geq 1$ by
definition and therefore $E_q > E$ is possible only when the covariation of $q_c$ and $N_c$ has a
*suppressing* effect, instead of enhancing. Once again, this result demonstrates the importance of
understanding the covariation of $q_c$ and $N_c$ for understanding the EF for autoconversion rate
parameterization.
Having looked at the observation-based EFs, we now check if the EFs derived based on
assumed PDFs (e.g., lognormal or bi-variate lognormal distributions) agree with the observation-
based results. As shown in Figure 11a, the $E_q$ based on Eq. ($6$) that assumes a lognormal
distribution for the subgrid variation of $q_c$ is in an excellent agreement with the observation-based
results. In contrast, the comparison is much worse for the $E_N$ in Figure 11b, which is not surprising
given the results from the July 18, 2017 case in Figure 6b. As one can see from Figure 5, the





marginal PDF of $N_c$ is often broad and sometimes even bimodal. The deviation of the observed
$N_c$ PDF from the lognormal distribution is probably the reason for the large difference of $E_N$ in
Figure 11b. As shown in Figure 11c, the $E$ derived based on Eq. (5) by assuming a bi-variate
lognormal function for the joint distribution of $q_c$ and $N_c$ tends to be larger than the observation-
based results. The reason for this overestimation is because the joint PDF of $q_c$ and $N_c$ is often
bimodal as seen in Figure 5. In such case, the small correlation coefficient $\rho$ due to the bimodality
is misinterpreted as a rather broad bi-variate lognormal distribution which in turn leads to an
overestimated $E$ value.
**6. Summary and Discussion**

In this study we derived the horizontal variations of $q_c$ and $N_c$, as well as their covariations

in MBL clouds based on the in-situ measurements from the recent ACE-ENA campaign and
investigated the implications for the EF of the autoconversion parameterization in the GCMs. The
main findings can be summarized as follows:
• In the July 18, 2017 case, the vertical variation of the mean values of $q_c$ and $N_c$ roughly

follows the adiabatic structure. The horizontal variances of $q_c$ and $N_c$ first decrease from

cloud base upward toward the middle of the cloud and then increase in the entrainment

zone. The correlation between of $q_c$ and $N_c$ generally increases from cloud base to cloud

top.

• In other selected cases, the horizontal variances of $q_c$ and $N_c$ show no statistically

significant dependence on the vertical height in cloud. However, the increasing trend of

the correlation between $q_c$ and $N_c$ from cloud base to cloud top remains robust.





• In a few selected "V" shape hlegs, the $q_c$ and $N_c$ follow a bimodal joint distribution
which leads to a poor linear correlation between them. The two modes in the bimodal
distribution correspond to the along-wind and cross-wind sides of the "V" shape hlegs.
• The observation-based physically complete $E$ that accounts for the covariation of $q_c$ and
$N_c$ has a robust decreasing trend from cloud base to cloud top, which can be explained
by the increasing trend of the $q_c$ and $N_c$ correlation from cloud base to cloud top.
• The $E$ estimated by assuming a monomodal bi-variate lognormal joint distribution
between $q_c$ and $N_c$ systematically overestimates the observation-based results,
especially for the hlegs with a bimodal $q_c$ and $N_c$ joint distribution. The omission of the
$N_c$ variation and its covariation with $q_c$ tends to lead to an overestimation of EF despite
the error cancellation.
These results provide the following two new understandings of the EF for the autoconversion
parameterization that have potentially important implications for GCM. First, our study indicates
that the physically complete $E$ has a robust decreasing trend from cloud base to cloud top. Because
the autoconversion process is most important at the cloud top, this vertical dependence of EF
should be taken into consideration in the GCM parametrization scheme. Second, our study
indicates that effect of the $q_c$ and $N_c$ correlation plays a critical role in determining the EF. Lately
a few novel modeling techniques have been developed to provide the coarse resolution GCMs
information of subgrid cloud variation, such as the PDF-based higher-order turbulence closure
method—Cloud Layer Unified By Binormals, CLUBB *(Golaz et al., 2002; Guo et al., 2015;*
*Larson et al., 2002)*. These models are able to provide parameterized subgrid variance of $q_c$ which
can be used in turn to estimate $E_q$. However, as shown in our study the $E_q$ tends to overestimate
the EF.



594   Our study has a couple of important limitations. First of all, our results are based on a

595 handful cases from a single field campaign. The lessons learned here need to be further examined

596 based on more data or tested in modeling studies. Second, we study provides only a

597 phenomenological analysis of the horizontal variations cloud microphysics in the MBL clouds and

598 the implications for the EF. Ongoing modeling research based on a comprehensive LES model is

599 being conducted to identify and elucidate the process-level physical mechanisms behind our

600 observational results.  Finally, this study is focused on the KK parameterization in estimating the

601 enhancement factors resulting from subgrid variability of $q_c$, $N_c$ and $q_c$-$N_c$ covariance. The

602 specific values are expected to differ when applied to other autoconversion parameterizations with

603 different power-law exponents.




**Acknowledgement:**
Z. Zhang acknowledges the financial support from the Atmospheric System Research (Grant DE-
SC0020057) funded by the Office of Biological and Environmental Research in the US DOE Office of
Science. The computations in this study were performed at the UMBC High Performance Computing
Facility (HPCF). The facility is supported by the U.S. National Science Foundation through the MRI
program (Grants CNS-0821258 and CNS-1228778) and the SCREMS program (Grant DMS-0821311),
with substantial support from UMBC. Co-author D. Mechem was supported by subcontract OFED0010-
01 from the University of Maryland Baltimore County and the U.S. Department of Energy's Atmospheric
Systems Research grant DE-SC0016522.





*Table 1 In situ cloud instruments from ACE-ENA campaign used in this study*

| Instruments | Measurements | Frequency | Resolution | Accuracy |
|---|---|---|---|---|
| **AIMMS** | P, T, RH, u,v,w | 20 Hz | / | / |
| **F-CDP** | DSD 2~50 µm | 10 Hz | 1 -2 µm | 2 µm |
| **2DS** | DSD 10 ~2500 µm | 1 Hz | 25 – 150 µm | 10 µm |

Table 2 *conditions of MBL sampled during the two IOPs of ACE-ENA campaign*

| Conditions Sampled | Research Flights | |
|---|---|---|
| | IOP1: June-July 2017 | IOP2: Jan.-Feb. 2018 |
| Mostly clear | 6/23, 6/29, 7/7 | 2/16 |
| Thin Stratus | 6/21, 6/25, 6/26, 6/28, 6/30, 7/4, 7/13 | 1/28, 2/1, 2/10, 2/12 |
| Solid StCu | 7/6, 7/8, 7/15 | 1/30 |
| Multi-layer StCu | 7/11, 7/12 | 1/24, 1/29, 2/8 |
| Drizzling StCu/Cu | 7/3, 7/17, 7/18, 7/19, 7/20 | 1/19, 1/21, 1/25, 1/26, 2/9, 2/11, 2/15, 2/18, 2/19 |


Table 3 A summary of selected RFs, and the selected hlegs and vlegs within each RF.

| Research Flight | Sampling pattern | Selected hlegs | Selected vlegs |
|---|---|---|---|
| July 18, 2017 IOP1 | "V" shape | 5, 6, 7, 8, 10, 11, 12 | 0, 1, 3 |
| July 20, 2017 IOP1 | "V" shape | 5, 6, 7, 8, 9, 13, 14 | 0, 1 |
| Jan. 19, 2018 IOP2 | "V" shape | 6, 7, 8, 15, 16 | 0, 1, 3 |
| Feb. 11, 2018 IOP2 | Straight-line | 4, 5, 6, 7, 12, 13 | 0, 1 |








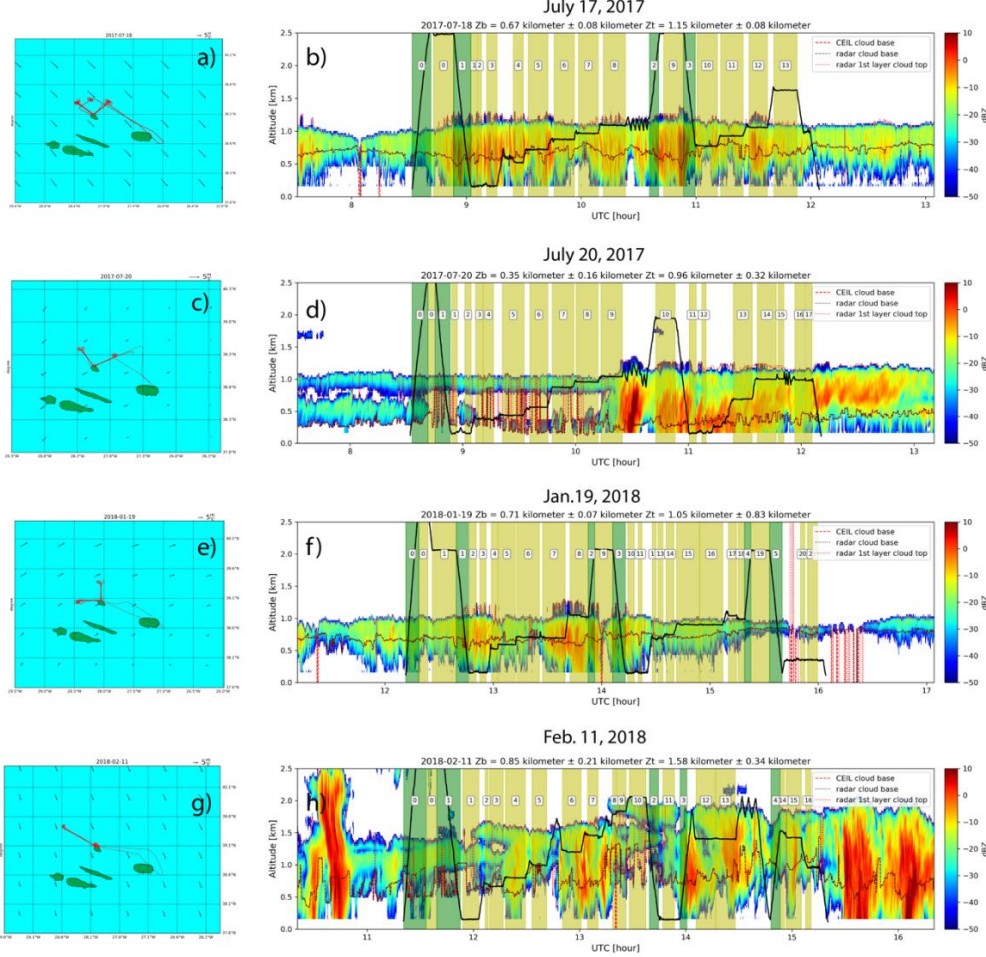


Figure 1 Four selected RF from the ACE-ENA for this study. (a) horizontal flight track of the G1 aircraft (red) during the July 18, 2017 RF. Small arrows in the figure indicate the wind vector at 900 mb. (b) vertical flight track of G1(thick black line) overlaid on the radar reflectivity contour by the ground-based KZAR. The dotted lines in the figure indicate the cloud base and top retrievals from ground-based radar and CEIL instruments. (c) and (d) same as (a) and (b), except for July 20, 2017 RF. (e) and (f) are for Jan. 19, 2018 RF. (g) and (h) are for Feb. 11, 2018 RF.







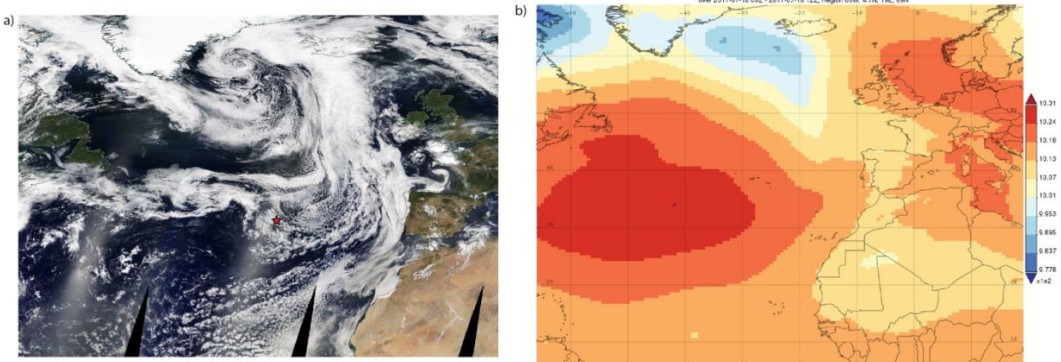


Figure 2 **(a)** The real color satellite image of the ENA region on July 18, 2017 from the MODIS.
The small red star marks the location of the ARM ENA site on the Graciosa Island; **(b)** The
averaged sea level pressure (SLP) of the ENA region on July 18, 2017 from the Merra-2
reanalysis.

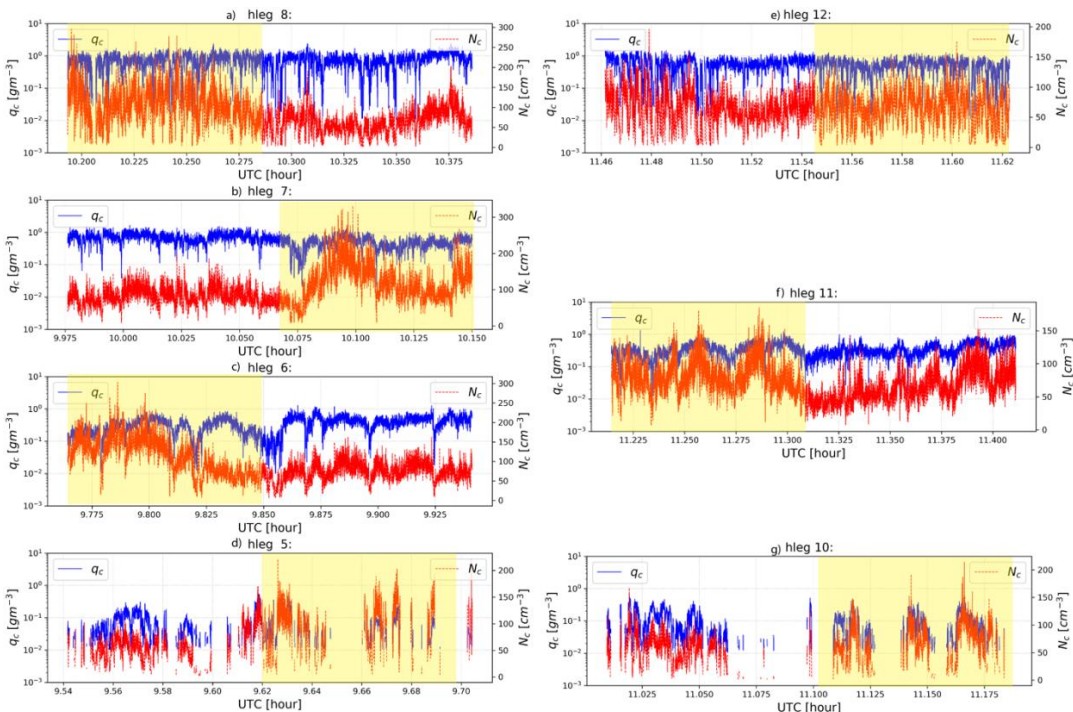


Figure 3 The horizontal variations of $q_c$ red) and $N_c$ (blue) for each selected hleg dervied from
the in situ FCDP instrument. The yellow-shaded time period in each plot corresponds to the
cross-wind side of the "V" shape flight track and the unshaded part corresponds to the along-
wind part. Note that plots are ordered such that the **(a)** hleg 8 and **(e)** hleg 12 are close to cloud
top; **(b)** hleg 6, **(c)** hleg 7 and **(f)** hleg 11 are sampled in the middle of clouds; **(d)** hleg 5 and **(g)**
hleg 10 are close to cloud base






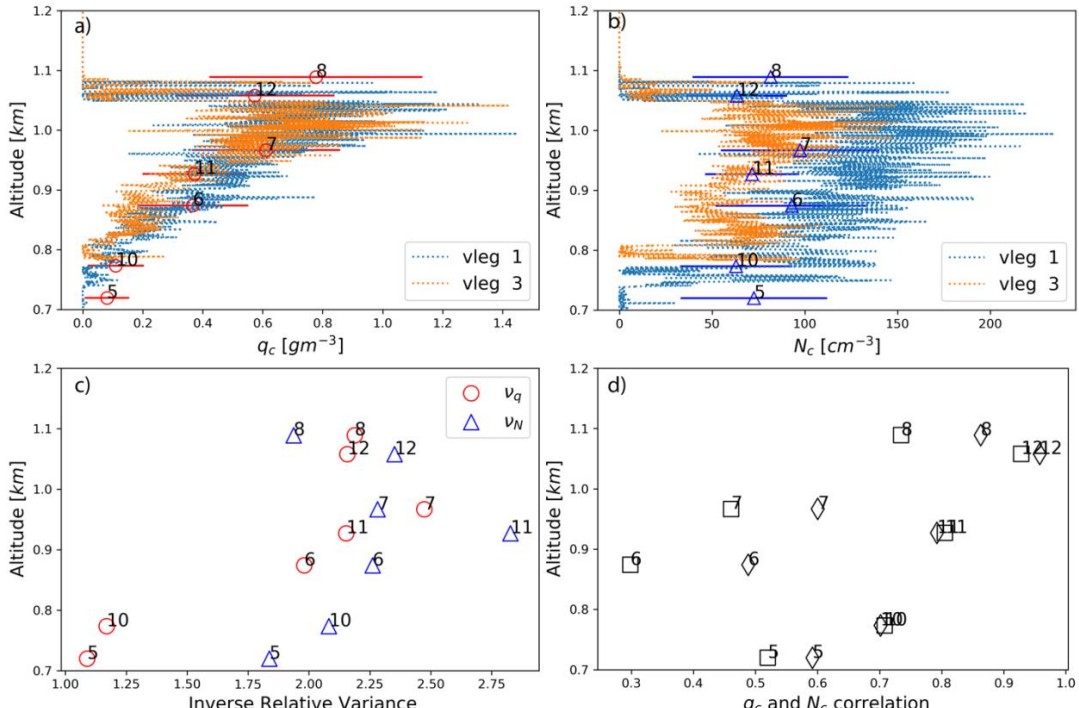


Figure 4 **(a)** The vertical profiles of $q_c$ derived from the vlegs (dotted lines) of the July 18, 2017
case. The overplotted red errorbars indicate the mean values and standard deviations of the $q_C$
derived from the selected hlegs at different vertical levels. **(b)** same as (a) except for $N_c$. **(c)** The
vertical profile of the inverse relative variances (i.e., mean divided by standard deivation) of $N_c$
(red circle) and $N_c$ (blue triangle ) derived from the hleg; **(d)** The vertical profile of the linear
correlation coefficienct between ln $(q_c)$ and ln$(N_c)$, i.e., $\rho_L$ (squre) and linear correlation
coefficienct between $q_c$ and $N_c$, i.e., $\rho$ (diamond).

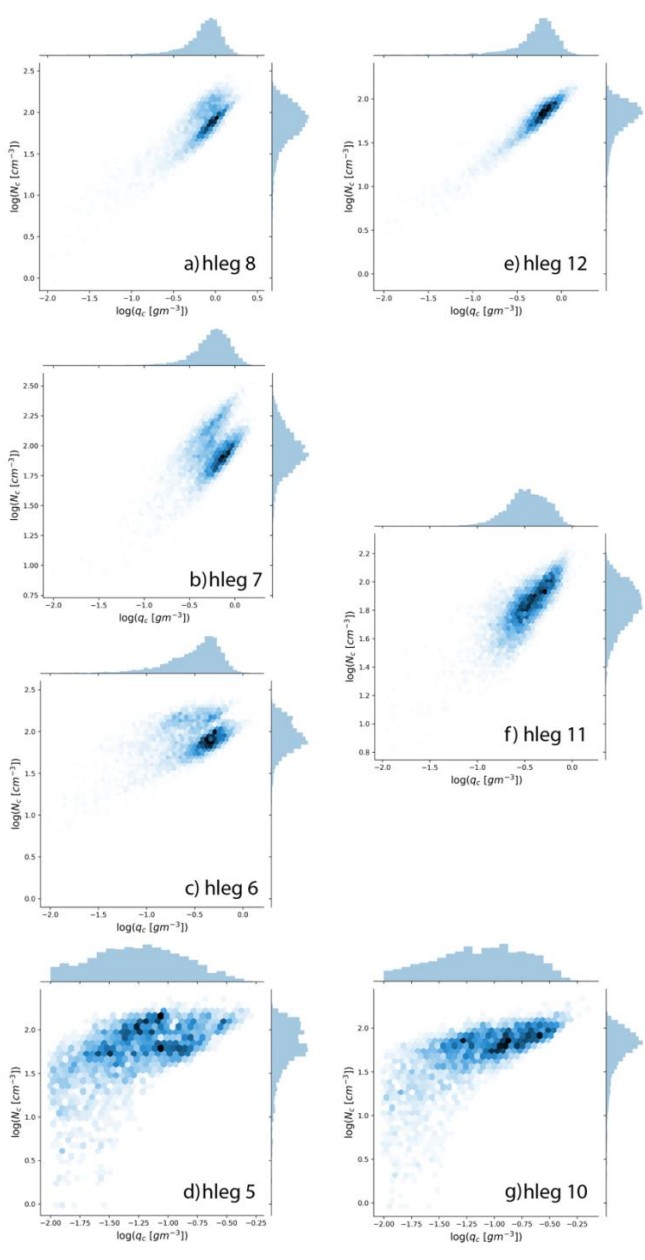


Figure 5 The joint distributions of the $q_c$ and $N_c$, along with the marginal histograms, for the 7
selected hleg from the July 18, 2017 RF. Same as Figure 3, the plots are ordered such that the **(a)**
hleg 8 and **(e)** hleg 12 are close to cloud top; **(b)** hleg 6, **(c)** hleg 7 and **(f)** hleg 11 are sampled in
the middle of clouds; **(d)** hleg 5 and **(g)** hleg 10 are close to cloud base.






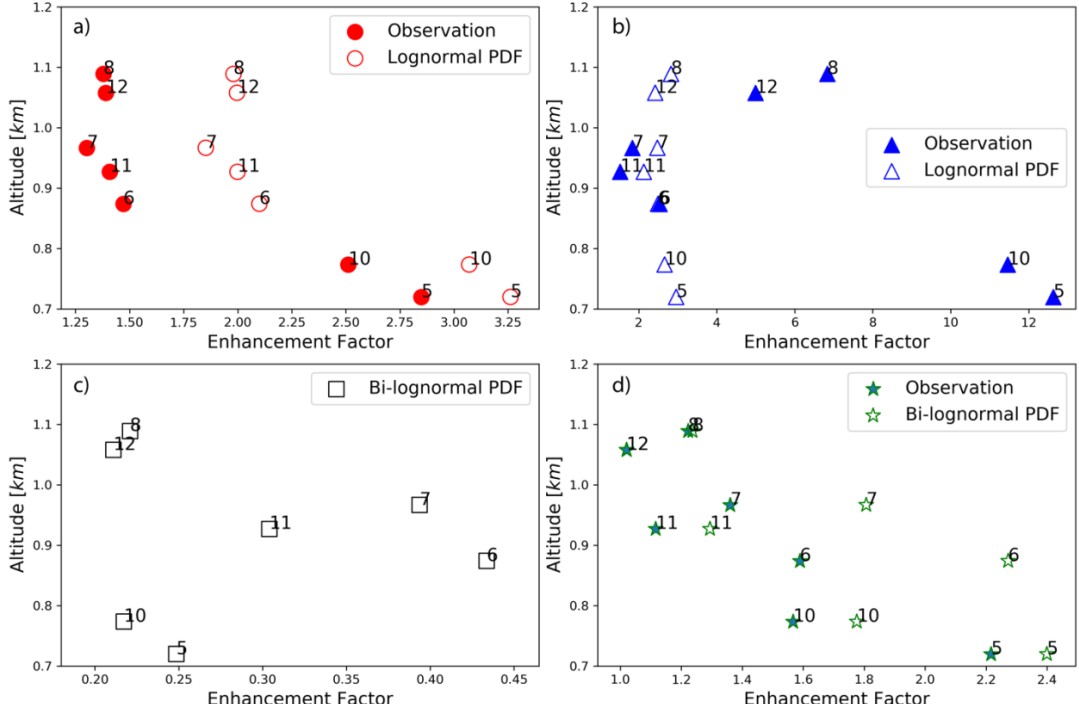


Figure 6 **(a)** $E_q$ as a function of height derived from observation based on Eq. (9) (solid circle)
and from the inverse relative variance $\nu_q$ assuming lognormal distribution based on Eq. (6) (open
circle). **(b)** $E_N$ as a function of height derived from observation based on Eq. (10) (solid triangle)
and from the inverse relative variance $\nu_N$ assuming lognormal distribution based on Eq. (7)
(open triangle). **(c)** $E_{COV}$ derived based on Eq. (8) as a function of height. **(d)** $E$ as a function of
height derived from observation based on Eq. (3) (solid star) and based on Eq. (5) assuming a bi-
lognormal distribution (open star). The numbers beside the symbols in the figure correspond to
the numbers of the 7 slected hlegs.





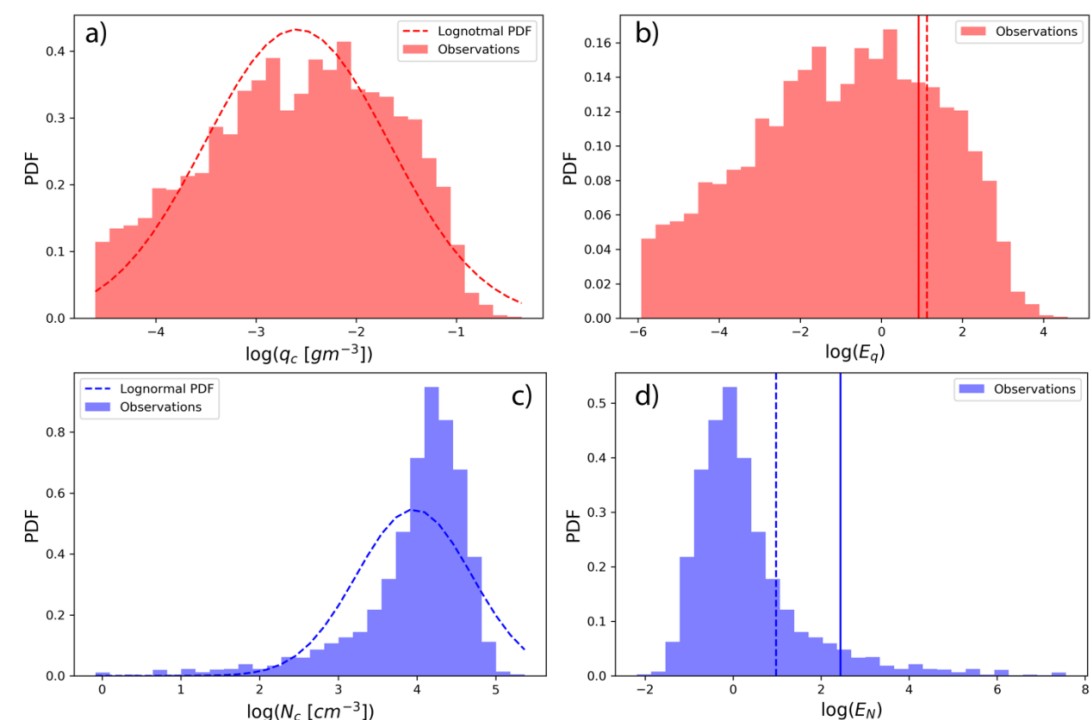


Figure 7 **(a)** Histogram of $\ln(q_c)$ based on observations from the hleg 10 (bars) and the lognormal
PDF (dashed line) based on the $\mu_{q_c}$ and $\sigma_{q_c}$ of hleg 10. **(b)** The histogram of $\ln(E_q)$ diagnosed
from the observed $q_c$ based on Eq. (9). The two vertical lines correspond to the leg-averaged
$\ln(E_q)$ derived based on the observed $q_c$ (solid) and the lognormal PDF (dashed line),
respectively. **(c)** Histogram of $\ln(N_c)$ based on observations from the hleg 10 (bars) and the
lognormal PDF (dashed line) based on the $\mu_{N_c}$ and $\sigma_{N_c}$ of hleg 10. **(d)** The histogram of $\ln(E_N)$
diagnosed from the observed $q_c$ based on Eq. (10). The two vertical lines correspond to the leg-
averaged $\ln(E_N)$ derived based on the observed $N_c$ (solid) and the lognormal PDF (dashed line),
respectively.




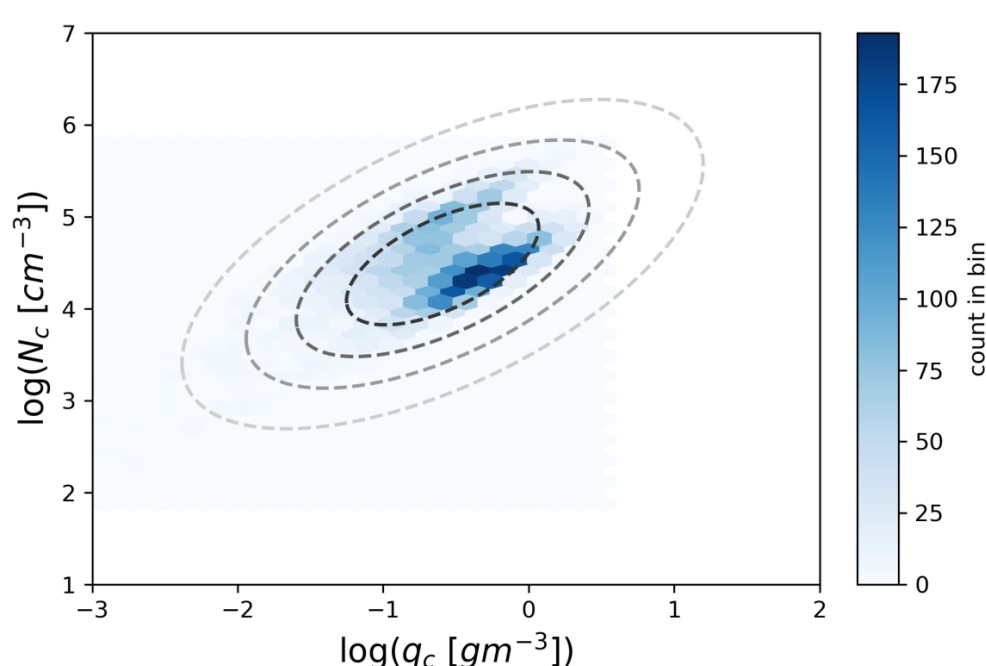


Figure 8 The joint PDF between $\ln(q_c)$ and $\ln(N_c)$ based on observations from hleg 7 (color
contour) in comparison with the bi-variate lognormal PDF (dashed line contour) which is derived
based on the $\mu_{q_c}$, $\sigma_{q_c}$, $\mu_{N_c}$ $\sigma_{N_c}$, and $\rho_L$ of hleg 7.









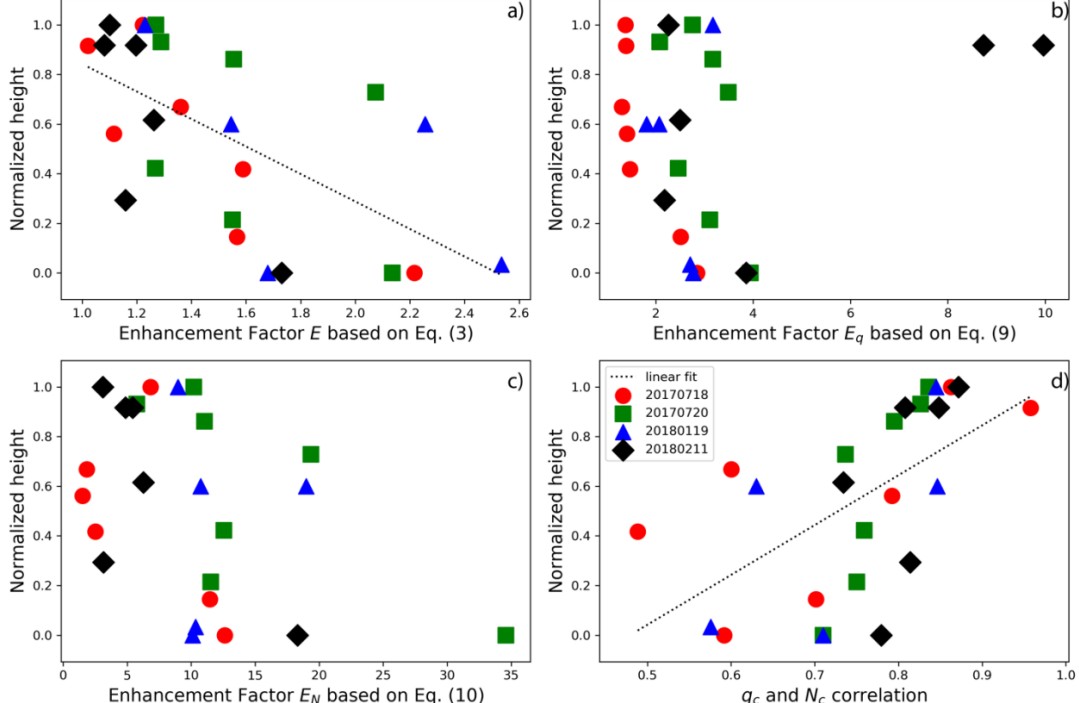


Figure 9 **(a)** The observation-based $E$ derived from Eq. (3) that accounts for the covariation of $q_c$ and $N_c$. **(b)** The observation-based $E_q$ derived from Eq. (9) that accounts for only the subgrid variation of $q_c$ **(c)** The observation-based $E_N$ derived from Eq. (10) that accounts for only the subgrid variation of $N_c$. **(d)** The correlation coefficient between $q_c$ and $N_c$. All quantities are plotted as a function of the normlized height $z^*_{hleg}$ in Eq. (11). The dashed lines correspond to a linear fit of the data when the fitting is statistically significant (i.e., P-value < 0.05).






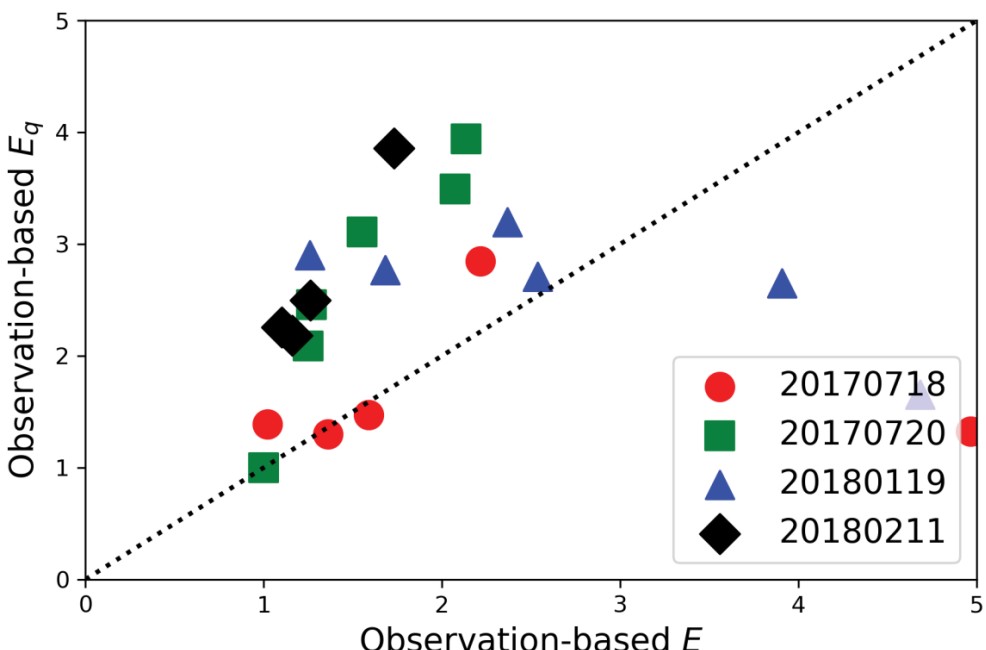


Figure 10 A comparison of observation-based $E$ and observation-based $E_q$ for all the selected
hlegs from all 4 selected RF.






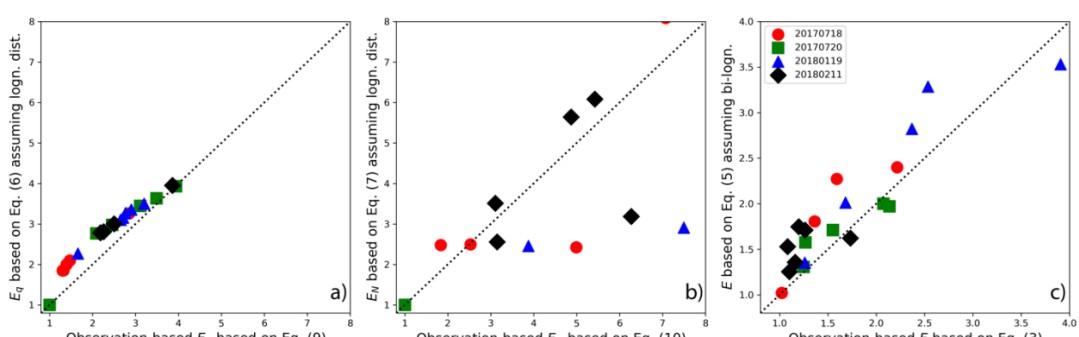


Figure 11 **(a)** A comparison of observation-based $E_q$ derived based on Eq. *(9)* and $E_q$ derived
based on Eq. *(6)* assuming lognormal distribution for subgrid $q_c$ observations for all the selected
hlegs. 12 **(b)** A comparison of observation-based $E_N$ derived based on Eq. *(10)* and $E_N$ derived
based on Eq. *(7)* assuming lognormal distribution or all the selected hlegs. **(c)** A comparison of
observation-based $E$ derived based on Eq. *(3)* and $E$ derived based on Eq. *(5)* assuming bi-
variate lognormal distribution for the subgrid joint distribution of $q_c$ and $N_c$.






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
