# Peer review of "Vertical Dependence of Horizontal Variation of Cloud Microphysics: Observations from the ACE-ENA field campaign and implications for warm rain simulation in climate models"

_Atmospheric Chemistry and Physics, 2020_

## Referee Comment (RC1) · Anonymous Referee #1 · 10 Sep 2020

This is an interesting and well written paper which discusses the variability of cloud water content, droplet number, and correlation between the two in stratiform clouds. The key findings seem to be the positive correlation between the water content and drop number, and the vertical dependence of this correlation, increasing towards cloud top. The consequences of these results for autoconversion parametrizations is discussed, showing that including the correlation is crucially important for getting the enhancement factor correct, although neglecting variation in drop number entirely is a surprisingly good estimate. I recommend publication after addressing some minor comments I've

listed below.

Minor comments

1. L45 and elsewhere - does the vertical dependence of EF really need to be accounted for, or would it suffice to simply get it right at cloud top? If the model is correctly representing the domination of accretion throughout the remainder of the cloud (e.g. as in Wood et al (2005b)), then autoconversion and the EF applied to it shouldn't matter so much here.

2. L103 - it would be worth noting here or later that some CMIP6 models (e.g. Walters et al, 2019, GMD) have adopted variable enhancement factors (better options!) based on the recent work cited, so it's not quite as bad as presented.

3. L157 - you could also note here that this suggests an equal amount of time should be dedicated to EFs for accretion, something which certainly isn't the case in the literature!

4. L271-273 - are your statistics for variance and correlation then only calculated in regions where qc>0.01g/m3, i.e. over the cloudy portion of the leg which may be <10km, or are they calculated over the entire >10km leg and include points where qc<0.01g/m3 (i.e. the zeros). As this is an important distinction, as it significantly affects the results (as shown in Witte et al, 2019, JAS) and also important for model developers to know to implement correctly in their microphysical parametrization (i.e. does the scheme work on in-cloud quantities or grid-box mean quantities).

5. Section 4.1 - how much do you think the results are affected by the non-stationarity of the cloud being sampled. i.e. for a model parametrization, you are interested in the variability at different heights within the cloud at the same time. But what you are sampling is the variability at different heights in the cloud at different times, up to several hours apart. The cloud is clearly evolving in this time, and how much might that evolution be affecting the results. For example, just looking at the variability in reflectivity in Figure 1 suggests there may be external factors affecting the cloud amount and drop

number, and currently this variability is being attributed to the height at which the flight leg during that period happened to be at.

6. Section 4.2 - do you have any hypothesis or theory why qc and Nc are positively correlated? Is there a physical mechanism for the correlation, or just something that happened to be the case for this study? One could imagine that areas with lower Nc will precipitate easier, thus lowering qc?

7. L491 - as with the comment above, do you think this really is a fortunate cancellation, or is there some underlying physical mechanism?

Typos

L41 - should say "effect that tends to make".

L60 - should probably say "e.g. Morrison and Gettleman" as theirs isn't the first or only scheme out there.

L538 - this should say "Eq is significantly larger than E" I think.

―――――――――――――――――――

---

## Referee Comment (RC2) · Anonymous Referee #2 · 20 Nov 2020

**General Comments**

This paper uses observational data from the ACE-ENA campaign to assess the horizontal variability and coverability of cloud water content and number concentration. The motivation for this study is the implication of these covariances on the parameterization of autoconversion in coarse resolution models. The study is unique in 2 regards: 1) it focuses on q-N covariability which is often ignored, and 2) it's evaluation of the coavaribilities as a function of cloud height. The study finds that the so-called enhancement factor for autoconversion decreases robustly from cloud base to cloud top

due to increasing correlation between q and N at cloud top. These results have important implications for the representation of unresolved cloud microphysical processes in climate and weather models.

I only have one critique of this paper. The authors should add non-precipitating clouds to the study. Once the clouds are drizzling the accretion process effectively dominates autoconversion in precipitation production, so in a sense we care more about the autoconversion process (and all of these covariabilities in non-precipitating clouds than we do in the precipitating clouds shown here. Also, there may be important differences between the covariability in non-precipitating and precipitating clouds and it would be informative to understand those differences if they exist.

The paper is very well written, adds to the field, and the methods are sound. I have some editorial comments below and a suggestion for future study.

In future studies (not in this paper) I would encourage the authors to look at height dependent correlations between qc and qr as they relate to accretion. Also understand in the height dependence of the precipitation fraction is critical in representing these unresolved processes.

Specific Comments:

None

Technical corrections:

Line 58: process -> processes

Line 370: explain -> explained

Figure 6: Can you put descriptive titles on each subplot or refer to the physical assumptions that correspond to each subplot in addition to referencing the equations to make it easier to figure out what everything means.

Line 485 abroad -> broad

Lines 537: Eq is used twice to mean two different things.

---

## Referee Comment (RC3) · Anonymous Referee #3 · 23 Nov 2020

Review of ACP Manuscript 2020GL087554

Title:

Vertical Dependence of Horizontal Variation of Cloud Microphysics: Observations from the ACE-ENA field campaign and implications for warm rain simulation in climate models

Authors:

Zhibo Zhang, Qianqian Song, David B. Mechem, Vincent E. Larson, Jian Wang, Yan-

gang Liu, Mikael K. Witte, Xiquan Dong, Peng Wu

Overview:

In addition to being interesting, I think this study is well thought out and the manuscript is well written. I found reading it to be a pleasure. In particular, I would like to compliment the authors on the quality of the introduction. That said, I do have one major concern (general comment #1, below) and several suggestions that might improve the manuscript.

Recommendation: Publish after major revisions

I have labeled this as a major revision, out of concern that responding to general comment #1 may require regenerating most of the figures in the manuscript and I want to give the authors ample time to do this, if necessary. This should not be taken to mean this is a poor work. I think it is excellent.

General Comments:

1) Bimodality in the relationship between qc and N

My only major concern with the analysis is that the bimodality in the relationship between qc and N has a significant impact on the results. The manuscript documents that this bimodality is associated with systematically different values for N measured when the aircraft was flying either along and across the wind. It seems to me that it is extremely unlikely that this would happen by random chance (that the aircraft just happened to observe a cloud field with two distinction regions with differing N that happen to align with the flight tracks) given that the same offset is found between legs occurring at different times and different altitudes. I can't think of any physical reason why this should occur. As such it seems to me that it is very likely that this is a measurement artifact. Accordingly:

(A) Discussion: Is it real?

Am I missing something simple? In general, I think the possible causes of this systematic shift need to be discussed.

(B) Subdivide the horizontal leg data between along and cross wind directions.

All of the analysis is based on analyzing full length of the horizontal legs, which if I understand correctly mean the outer scale is about 60 km. That is, the enhancement factors you present represent the enhancement in going from the ∼10 m scale of the FCDP measurement to a nominal grid scale of ∼60 km.

This is a choice you made. There is (as far as I can see) nothing that prevents you from dividing your hlegs into separate along and cross wind legs (I presume each will be about 30 km) and instead examine the variability on this smaller scale ∼ 30 km scale.

Obviously doing this will increase the sampling uncertainty associated with each leg, but you will have twice the number of points in most of your figures and more critically it will significantly reduce the impact of any measurement bias in N associated with the along or cross wind measurements.

2) Flying in and out of cloud?

How are data for those legs where the aircraft goes in and out of cloud being handled? I assume that you didn't include a lot of zeros in either qc and/or N when calculating the means and variances, or for that matter, the directly calculating E values (e.g., equation 9)? Please describe what was done and comment on the impact this might have on the results.

3) Inner Scale Dependence

One of the most interesting points in the paper is that the PDF of N is not lognormally distributed near the boundaries. It seems to me that this is likely due to entrainment preferentially evaporating the smallest cloud droplets first. The time series is very spikey in nature (spikes having low N values). This suggests to me that this effect

might be greatly reduced if you started from a larger inner scale (100 or 200 m) rather than the $\sim$ 10 m.

I think it would be a nice addition to your analysis to first average the FCDP data to a scale of 100 to 200 m (use 1 Hz data) and examine how this impacts the skewness of the distributions and the ability of the bivariate log normal to model the variability and enhancement factors.

4) Dependency rather than trend

I think the discussion is overly focused on whether there is a (linear) trend in the data. Let's look at Figure 9. To me, "trend" just doesn't seem like the right word to describe the situation in Figure 9a. I suggest a more apt description might be, "Figure 9a shows significant scatter in E with altitude, with smaller values and relatively little scatter near cloud top, and decidedly larger values near cloud base."

I understand entirely that the change with height projects on to a line and the correlation is significant. There is a vertical dependence. But it is not obvious to me that there is a linear dependence or trend.

Importantly I see Figure 9c has this same pattern, with larger En values (on average) at the bottom than the top, as does 9b for Eq with the exception of the two outliers near cloud top.

Somewhat similarly in 9d, the correlations are clearly stronger and there is less variability near cloud top. But it is not clear to me there is a linear dependence. I note the bimodal points are causing much of the variability mid-cloud.

I recognize this is partly philosophical, but I think it would be better to recast some of the discussion in terms of a there being a vertical dependence rather than a "trend", though perhaps with more points (general comment 1b) a trend will become clearer.

Specific Comments:

Line 213. What is the WCM-2000? Please provide a reference and perhaps add to table #1. Perhaps add a figure demonstrating agreement in qc, or at least quantify what "excellent" agreement means (e.g. Bias less than X and RMS difference less than Y% at 10 Hz?). I note here that this study is about variability and so ideally it would be good to establish variability in qc is the same from both sensor (not simply that the bias between two measurements is small).

Table #1. Perhaps change label "Accuracy" to "Particle Size Intervals" or something similar?

Line 214. A value of 20 microns for ðİŚ§* is about as small a threshold as might be chosen, and there are often a significant number of particles in the 20 to 50 micron size range for clouds that are described as "non precipitating". I expect particle concentrations in the 20 to 50 micron size range will co-vary with the concentration of particles less than 20, so I am not surprised that you see little sensitivity. Nonetheless I think it would have been better test of the sensitivity by choosing values for r* of 20 and 50, rather than 20 +/-5. In particular, you show later in the manuscript that N (with the cut off of r* of 20) is not well described by a lognormal near cloud-top or cloud-base. Is this result sensitive to the choice of r*?

Line 238. Do I understand correctly that only flight labeled as drizzling in table #2 were considered? If so, how were the groupings in Table #2 established? Does "non precipitating" in table #2 mean only particles less than 20 microns where present OR Only when drizzle is clearly falling from clouds (from radar and "pilots" on line 269)? Perhaps it doesn't matter for this paper, but I presume models will apply autoconversion rate parameterizations all the time and I think it would be useful to examine precipitation formation for cloud without obvious precipitation / virga falling from the bottom OR otherwise clearly establish the conditions under which your results apply.

Line 269. I don't fully understand this criteria or how it is being applied. Is this requirement applied to each horizontal leg or just for choosing cases (see comment for line

238)? If individual legs, are you focusing on the portion of the leg that occurs near the radar? Is there a reflectivity threshold? In general, you have CDP and 2DS data and so I don't understand the choice to rely on radar or pilots.

Line 274. Does "same region" here mean the hlegs must occur along roughly the same track, that is the same set of latitudes and longitudes? If yes, I don't understand the rationale for this. As long as you are sampling the same cloud field (i.e. it is reasonable to expect a representative measure of the variability) and you are using hlegs with the same total length, why do you care if the hlegs and nominally stacked?

Line 332. Previous studies? Please provide a reference.

Line 443. "slight"? It is not clear to me whether this difference would or would not have a significant impact on a model simulation. Perhaps just indicate the bivariate lognormal results in an EF value that is about 0.5 larger (be quantitative rather than qualitative).

Line 541. It seems natural to expect qc and N will be positively correlated, since stronger updraft mean both activating more droplets and condensing more water.

Line 503-12. The use of a "relative hleg altitude" seem to me to be "just as prone" or even "more prone" to misinterpretation as using cloud boundaries from either radar/lidar or the vlegs. Here you are effectively demanding that the highest and lowest legs in each flight should have the same relationship regardless of how close or far they are from the actual cloud top or cloud base. I guess the question is, does it matter? Based on your 4th criteria (line 281) I would hope not.

Minor Comments

Line 55-6. Replace "... at the spatial scales much smaller .. " with "... on spatial scales that are much smaller ..."

Line 56-7. Remove the phrase " ... making the simulation of these processes in GCMs highly challenging." This is entirely redundant with the "challenging" remark on the

previous sentence.

Line 164. Perhaps rephrase as "... better understand the horizontal variations in ðĺŚððĺŚŘ and ðĺŚĄðĺŚŘ, their covariation, and the dependence on the vertical height in cloud ..."

Line 184. Replace "seasonable" with "seasonal"

Line 185. Perhaps add comma and remove "and" so the sentence reads " ... the MBL (Dong et al., 2014; Rémillard 185 et al., 2012), to improving cloud parameterizations in the GCMs (Zheng et al., 2016), to validating the space-borne remote sensing products of MBL clouds (Zhang et al., 2017)."

Line 189. Change "is" to "was", and perhaps simplify to read "In 2013 a permanent measurement site was established by the ARM program on Graciosa Island, and is typically referred to as the ENA site (Voyles and Mather, 2013)."

Figure 1. I presume the green blobs are islands? Perhaps explain this in the caption along with note about location of the ARM site and likewise describe what the number in white boxes and colored stripes mean in radar panels? Also I suggest higher resolution figure would be helpful here.

Figure 9. The solid symbols seem a bit problematic here. I am pretty sure that in panel c, there is a red dot near cloud top that must be hidden under the other symbols – and this make me wonder if other symbols might also be hidden.

Line 537. I think you mean "Eq is larger than E", and you need to change the second occurrence of Eq in this sentence to be simply E.

Line 561. Perhaps change to read "... implications of subgrid variability as relates to the enhancement of autoconversion rates ..."

Line 566. Perhaps add "near cloud top" to the end of the sentence. I think it is reasonable to make the association with cloud-top entrainment but obviously, the study

doesn't define entrainment zone as best I recall.

Line 569. Change "we" to "our".
* * *

---

## Author Response (AR1)

Responses to the reviewer #1:

*This is an interesting and well written paper which discusses the variability of cloud water content, droplet number, and correlation between the two in stratiform clouds. The key findings seem to be the positive correlation between the water content and drop number, and the vertical dependence of this correlation, increasing towards cloud top.*

*The consequences of these results for autoconversion parametrizations is discussed, showing that including the correlation is crucially important for getting the enhancement factor correct, although neglecting variation in drop number entirely is a surprisingly good estimate. I recommend publication after addressing some minor comments I've listed below.*

**Reply**: We thank this reviewer for the encouraging, insightful and constructive comments, which really help improve the manuscript significantly.

Before addressing your comments/questions below, first we would like to provide a summary of the major revisions made to the manuscript:
- We revised significantly the part about the bimodal joint distribution between qc and Nc in section 4. In particular, we pointed out that it is most likely just a coincidence that each side of the "V" shape track sampled one mode of the bimodal distribution. The along/across wind difference between the two sides is unlikely to be the cause of the bimodality.
- Three new cases that are either non-precipitating or weakly precipitating were added to the paper and they have no overall impacts on the conclusions. The flight track and radar reflectivity plots for all the cases, except for July 18, 2017, are provided in the supplementary material.
- A small bug in our code was found and fixed. This bug affects the computation of the EF based on lognormal distributions. As a result, the $E_q$ based on the lognormal PDF agrees very well with the observation-based $E_q$ (new Figure 6a), and the $E$ based on the bivariate lognormal distribution agrees well with the observation-based $E$ (new Figure 6d). Because of this, the Figure 8 was removed from the paper.
- Most figures are revised/updated per request/suggestion of the reviewers.

After these revisions, we think the paper is much improved and more focused, although the general conclusions still hold.

*Minor comments*

*1. L45 and elsewhere - does the vertical dependence of EF really need to be accounted for, or would it suffice to simply get it right at cloud top? If the model is correctly representing the domination of accretion throughout the remainder of the cloud (e.g. as in Wood et al (2005b)), then autoconversion and the EF applied to it shouldn't matter so much here.*

**Reply**: Indeed, the autoconversion and accretion have different relative importance at different vertical locations of the cloud. As we pointed out in the paper, most previous studies have ignored the vertical dependence of the EF and the consequential impacts on autoconversion rate simulation. In fact, this is the major motivation of this study. Based on observations, our study reveals that the EF at cloud top is significantly different (smaller) than the lower parts of the clouds and we also explain the underlying physics. Without a good understanding of the "truth" and underlying physics, how could we make sure that the model "*gets it right at cloud top*" and also "*correctly representing the domination of accretion throughout the remainder of the cloud*"?

*2. L103 - it would be worth noting here or later that some CMIP6 models (e.g. Walters et al, 2019, GMD) have adopted variable enhancement factors (better options!) based on the recent work cited, so it's not quite as bad as presented.*

**Reply**: Good point. We revised the paper to point out that the latest generation of GCMs may have adopted more advanced schemes to account for the EF than using a constant EF (around Line 141 in the revised manuscript). On the other hand, it is also important to note that Walters et al (2019) adopted the cloud-regime dependent and scale-aware schemes developed by Hill et al. (2015) and Boutle et al. (2014) to account for subgrid cloud variability. However, even these advanced schemes only consider the subgrid variability of qc only but ignore the variability of Nc and its co-variability with qc. Therefore, they also have important limitations.

*3. L157 - you could also note here that this suggests an equal amount of time should be dedicated to EFs for accretion, something which certainly isn't the case in the literature!*

**Reply**: Good point. We noted here that the vertical dependence is important for both autoconversion and accretion.

*4. L271-273 - are your statistics for variance and correlation then only calculated in regions where qc>0.01g/m3, i.e. over the cloudy portion of the leg which may be <10km, or are they calculated over the entire >10km leg and include points where qc<0.01g/m3 (i.e. the zeros). As this is an important distinction, as it significantly affects the results (as shown in Witte et al, 2019, JAS) and also important for model developers to know to implement correctly in their microphysical parametrization (i.e. does the scheme work on in-cloud quantities or grid-box mean quantities).*

**Reply**: indeed, this is important. All the analyses are based on in-cloud observations (e.g., regions with qc>0.01g/m3). We pointed this out as suggested.

*5. Section 4.1 - how much do you think the results are affected by the non-stationarity of the cloud being sampled. i.e. for a model parametrization, you are interested in the variability at different heights within the cloud at the same time. But what you are sampling is the variability at different heights in the cloud at different times, up to several hours apart. The cloud is clearly evolving in this time, and how much might that evolution be affecting the results. For example, just looking at the variability in reflectivity in Figure 1 suggests there may be external factors affecting the cloud amount and drop number, and currently this variability is being attributed to the height at which the flight leg during that period happened to be at.*

**Reply**: This is a very good question! It is certainty possible that the selected clouds in this study are "non-stationary". But it has to be noted that we observed similar vertical variations of qc and Nc in all four selected cases. It seems highly unlikely that the temporal evaluations of the clouds in all four cases conspire to confound our results in the same way. Based on this consideration, we assume that the temporal evolution of clouds is an uncertainty that could lead to random errors but not the overall conclusions. Of course, it is extremely difficult, if not impossible, to address this issue using air-borne in situ measurements alone due to their inherent limitations as you pointed out. Ground-based radars can provide a reasonable estimate of the vertical profile of qc at the temporal-resolution of 5-mintues or so. But currently there is no reliable retrieval of the vertical profile of Nc from ground radar, yet. The only useful tool in this regard is model simulation. In fact, we are currently simulating the July 18, 2017 case using a LES model which will hopefully help us understand both the spatial and temporal evolution of the clouds and thereby subgrid variability. But this is beyond the scope of this paper. Nevertheless, we pointed out this important limitation at section 4.1 and also at the end of the paper along with other limitations of this study.

*6. Section 4.2 - do you have any hypothesis or theory why qc and Nc are positively correlated? Is there a physical mechanism for the correlation, or just something that happened to be the case for this study? One could imagine that areas with lower Nc will precipitate easier, thus lowering qc?*

**Reply**: We do think there is some underlying physical processes that could lead to the positive correlation between qc and Nc at cloud top. As you pointed out, one possibility is the autoconversion process itself that converts the cloud water to rainwater and at the same time reducing the Nc. Another important possibility is the inhomogeneous mixing as a result of cloud top entrainment, which reduces the qc and Nc simultaneously leading to positive correlation between them. We pointed out these possibilities in the revised manuscript.

*7. L491 - as with the comment above, do you think this really is a fortunate cancellation, or is there some underlying physical mechanism?*

**Reply**: As mentioned above, we do believe there is some underlying physical processes that could lead to the positive correlation between qc and Nc at cloud top. As discussed in Eq. (5), because the qc and Nc is positively correlated the Ecov term is smaller than unity. In contrast, the En and Eq terms are always larger than unity. Therefore, it is expected (and explained in the paper) that the Ecov term tends to cancel the effects of En and Eq. But to what extent Ecov and En terms cancel out one another depends quantitively on the variability of Nc and its correlation with qc. But a more important point we would argue is that one should not rely on "fortunate cancellation". It would be more robust and physically sound to take all three terms, i.e., Eq, En and Ecov, into consideration.

*Typos*
*L41 - should say "effect that tends to make".*
*L60 - should probably say "e.g. Morrison and Gettleman" as theirs isn't the first or only scheme out there.*
*L538 - this should say "Eq is significantly larger than E" I think.*

**Reply**: these typos are all corrected as suggested. Thanks for pointing them out!

Responses to the reviewer #2:

General Comments

*This paper uses observational data from the ACE-ENA campaign to assess the horizontal variability and coverability of cloud water content and number concentration. The motivation for this study is the implication of these covariances on the parameterization of autoconversion in coarse resolution models. The study is unique in 2 regards: 1) it focuses on q-N covariability which is often ignored, and 2) it's evaluation of the coavaribilities as a function of cloud height. The study finds that the so-called enhancement factor for autoconversion decreases robustly from cloud base to cloud top due to increasing correlation between q and N at cloud top. These results have important implications for the representation of unresolved cloud microphysical processes in climate and weather models.*

**Reply**: We thank this reviewer for the encouraging, insightful and constructive comments, which really help improve the manuscript significantly.

Before addressing your comments/questions below, first we would like to provide a summary of the major revisions made to the manuscript:

- We revised significantly the part about the bimodal joint distribution between qc and Nc in section 4. In particular, we pointed out that it is most likely just a coincidence that each side of the "V" shape track sampled one mode of the bimodal distribution. The along/across wind difference between the two sides is unlikely to be the cause of the bimodality.
- Three new cases that are either non-precipitating or weakly precipitating were added to the paper and they have no overall impacts on the conclusions. The flight track and radar reflectivity plots for all the cases, except for July 18, 2017, are provided in the supplementary material.
- A small bug in our code was found and fixed. This bug affects the computation of the EF based on lognormal distributions. As a result, the $E_q$ based on the lognormal PDF agrees very well with the observation-based $E_q$ (new Figure 6a), and the $E$ based on the bivariate lognormal distribution agrees well with the observation-based $E$ (new Figure 6d). Because of this, the Figure 8 was removed from the paper.
- Most figures are revised/updated per request/suggestion of the reviewers.

After these revisions, we think the paper is much improved and more focused, although the general conclusions still hold.

*I only have one critique of this paper. The authors should add non-precipitating clouds to the study. Once the clouds are drizzling the accretion process effectively dominates autoconversion in precipitation production, so in a sense we care more about the autoconversion process (and all of these covariabilities in non-precipitating clouds than we do in the precipitating clouds shown here. Also, there may be important differences between the covariability in non-precipitating and precipitating clouds and it would be informative to understand those differences if they exist.*

**Reply**: Thanks for the suggestion. Indeed, in the original manuscript, we selected only 4 heavily drizzling cases with strong radar reflectivity and precipitation reaching the ground. We didn't select the non-precipitating cloud cases for a couple of reasons. The first reason is to ensure that autoconversion and accretion processes are active in the selected case. The relevance of an enhancement factor for a cloud not producing precipitation is nebulous. The second reason is more practical. It is because non-precipitating clouds are usually physically thinner than precipitating clouds, which makes it difficult for the airplane to sample different vertical locations of the clouds. As a result, there is often only one or two in-cloud hlegs for the non-precipitating clouds.

Nevertheless, per your suggestion, we selected three non-precipitating or weakly precipitating cases: 1) 2017-07-13 (non-precipitating) 2) 2018-01-26 (weakly precipitating at cloud base but no perception on the ground) 3) 2018-02-07 (very weakly precipitating at cloud base but no perception on the ground) and added them to the revised manuscript. The radar reflectivity curtain with vertical flight track for these three cases are shown in Figure 1 below. The abovementioned challenge of sampling thin non-precipitating cloud can be clearly seen in Figure 1a for the 2017-07-13 case. The selected hlegs and vlegs for these cases are summarized in Table 1. We repeated the same analyses for these new cases as for other cases, i.e., the vertical and horizontal structures of qc and Nc, as well as the EF, for these newly added cases. Overall, the results from these newly added non-precipitating cloud cases are highly similar to those based on the July 18, 2017 case as discussed in section 4. Take the 2018-02-07 case for example. Figure 2 shows the vertical variation of the inverse relative variances $v_q$ and $v_N$. Apparently, both $v_q$ and $v_N$ demonstrate a pattern similar to that of the July-18, 2017 case (see Figure 4c of the paper), i.e., increasing first from cloud base (hleg 1 -> hleg 2) and then decrease toward cloud top (hleg 3). Therefore, these newly added cases do not affect the general conclusion although they add to the statistics.

[Figure]

*Figure 1 Three non-precipitating (or weakly precipitating) clouds added to the revised manuscript.*

*Table 1 A summary of selected RFs, and the selected hlegs and vlegs within each RF. (newly added non-precipitating cases are highlighted in bold font)*

| Research Flight | Precipitation | Sampling pattern | Selected hlegs | Selected vlegs |
|---|---|---|---|---|
| **July 13, 2017** | **Non- Precipitating** | **Straight-line** | **3, 4, 5** | **0, 1, 3** |
| July 18, 2017 | Precipitation reaching ground | "V" shape | 5, 6, 7, 8, 10, 11, 12 | 0, 1, 3 |
| Jan. 19, 2018 | Precipitation reaching ground | "V" shape | 6, 7, 8, 15, 16 | 0, 1, 3 |
| July 20, 2017 | Precipitation reaching ground | "V" shape | 5, 6, 7, 8, 9, 13, 14 | 0, 1 |
| **Jan. 26, 2018** | **Precipitation only at cloud base** | **Straight-line** | **3, 4, 5, 9, 10, 11** | **0, 1, 3** |
| **Feb. 07, 2018** | **Non- Precipitating** | **"V" shape** | **1, 2, 3, 5** | **0, 1** |
| Feb. 11, 2018 | Precipitation reaching ground | Straight-line | 4, 5, 6, 7, 12, 13 | 0, 1 |

[Figure]

*Figure 2 vertical dependence of the inverse relative variances for the Feb. 07 2018 case.*

*The paper is very well written, adds to the field, and the methods are sound. I haves some editorial comments below and a suggestion for future study. In future studies (not in this paper) I would encourage the authors to look at height dependent correlations between qc and qr as they relate to accretion. Also understand in the height dependence of the precipitation fraction is critical in representing these unresolved processes.*

Specific Comments:

*None*

*Technical corrections:*

*Line 58: process -> processes*

**Reply**: corrected

*Line 370: explain -> explained*

**Reply**: corrected

*Figure 6: Can you put descriptive titles on each subplot or refer to the physical assumptions that correspond to each subplot in addition to referencing the equations to make it easier to figure out what everything means.*

**Reply**: Good suggestion. We added titles to each subplot of Figure 6 and some other figures.

*Line 485 abroad -> broad*

**Reply**: corrected

*Lines 537: Eq is used twice to mean two different things.*

**Reply**: The second $E_q$ should be $E$

Responses to the reviewer #3

*Overview:*
*In addition to being interesting, I think this study is well thought out and the manuscript is well written. I found reading it to be a pleasure. In particular, I would like to compliment the authors on the quality of the introduction. That said, I do have one major concern (general comment #1, below) and several suggestions that might improve the manuscript.*

*Recommendation: Publish after major revisions*

*I have labeled this as a major revision, out of concern that responding to general comment #1 may require regenerating most of the figures in the manuscript and I want to give the authors ample time to do this, if necessary. This should not be taken to mean this is a poor work. I think it is excellent.*

**Reply**: We thank this reviewer for the encouraging, insightful and constructive comments, which really help improve the manuscript significantly.

Before addressing your comments/questions below, first we would like to provide a summary of the major revisions made to the manuscript:
- We revised significantly the part about the bimodal joint distribution between qc and Nc in section 4. In particular, we pointed out that it is most likely just a coincidence that each side of the "V" shape track sampled one mode of the bimodal distribution. The along/across wind difference between the two sides is unlikely to be the cause of the bimodality.
- Three new cases that are either non-precipitating or weakly precipitating were added to the paper and they have no overall impacts on the conclusions. The flight track and radar reflectivity plots for all the cases, except for July 18, 2017, are provided in the supplementary material.
- A small bug in our code was found and fixed. This bug affects the computation of the EF based on lognormal distributions. As a result, the $E_q$ based on the lognormal PDF agrees very well with the observation-based $E_q$ (new Figure 6a), and the $E$ based on the bivariate lognormal distribution agrees well with the observation-based $E$ (new Figure 6d). Because of this, the Figure 8 was removed from the paper.
- Most figures are revised/updated per request/suggestion of the reviewers.

After these revisions, we think the paper is much improved and more focused, although the general conclusions still hold.

*General Comments:*
*1) Bimodality in the relationship between qc and N*
*My only major concern with the analysis is that the bimodality in the relationship between qc and Nc has a significant impact on the results. The manuscript documents that this bimodality is associated with systematically different values for N measured when the aircraft was flying either along and across the wind. It seems to me that it is extremely unlikely that this would happen by random chance (that the aircraft just happened to observe a cloud field with two distinction regions with differing N that happen to align with the flight tracks) given that the same offset is found between legs occurring at different times and different altitudes. I can't think*

*of any physical reason why this should occur. As such it seems to me that it is very likely that this is a measurement artifact. Accordingly:*

**Reply**: Indeed, this is a very interesting and puzzling phenomenon/observation. Although we believe the bimodality is "real" (in other words, it is unlikely an instrument artifact), we think it is most likely just a coincidence that each mode is sampled by one side of the "V" shape track. We do NOT think that the difference in wind pattern (i.e., along wind vs. across wind) "causes" bimodality. Instead, the two sides of the "v"-shaped happen to sample different regions of the "virtual" grid box that have different qc-Nc relations. One can imagine that, if the airplane can sample every point in the "virtual" grid box, then there would be no along-wind vs. across-wind (or from a different perspective, all points can be either along wind or across wind). So, there is no causal relation between along/across-wind difference and the bimodality. They are both results of "v"-shaped sampling pattern. On the other hand, it is important to note that a bimodal sub-grid joint PDF between qc and Nc can very well exist in the nature, which could be a result of sub-grid variation of, for example, vertical draft velocity, precipitation and/or CCN.

In short, we argue that the bimodal joint PDF of qc and Nc is "real", but it is not a result of the along/across wind pattern difference. We will further elaborate on this point in the context of your questions/comments. We have also revised the manuscript accordingly.

*(A) Discussion: Is it real?*
*Am I missing something simple? In general, I think the possible causes of this systematic shift need to be discussed.*
**Reply**: As we explained above, we do believe that the observed bimodal joint PDF between qc and Nc is "real". Perhaps, we should not have emphasized the along-wind vs. across wind difference between the two sides of the v shape flight, as it is most likely just a coincidence.

In the revised the manuscript, we point it out that the along/across wind difference and the bimodal joint PDF are likely coincidence. We also pointed it out that some sub-grid variations, for example, vertical draft velocity, precipitation and/or CCN can lead to bimodal joint PDF between qc and Nc.

*(B) Subdivide the horizontal leg data between along and cross wind directions. All of the analysis is based on analyzing full length of the horizontal legs, which if I understand correctly mean the outer scale is about 60 km. That is, the enhancement factors you present represent the enhancement in going from the ~10 m scale of the FCDP measurement to a nominal grid scale of ~60 km. This is a choice you made. There is (as far as I can see) nothing that prevents you from dividing your hlegs into separate along and cross wind legs (I presume each will be about 30 km) and instead examine the variability on this smaller scale ~30 km scale. Obviously doing this will increase the sampling uncertainty associated with each leg, but you will have twice the number of points in most of your figures and more critically it will significantly reduce the impact of any measurement bias in N associated with the along or cross wind measurements.*
**Reply**: Yes, your understanding is correct. We are using the high-resolution (~10 m) FCDP measurements to investigate the sub-grid scale variations of qc and Nc for a nominal GCM grid of 60 km. We made the choice *not* to separate the two sides of the "v" shape flight because the current generation GCMs usually have a typical resolution of ~100km. A ~60 km grid box would be more relevant to climate modeling in this regard.

Before we submitted the manuscript, we had already done the sensitivity study as your suggested, i.e., separating the two sides of "v" shape flight for the July 18, 2017 case. The new hlegs after the resampling are shown in Figure 1 below. We selected the new hlegs #6 to #13 (corresponding to the original hleg #5 to #8) and #16 to #21 (corresponding to the original hleg #10 to #12) for further analysis. As listed in Table 1, the new hlegs 6, 9, 10, 13, 16, 19, 20 are from the west side of the "v" shape flight (along-wind) and 7, 8, 11, 12, 17, 18, 21 are from the east side of the "v" shape flight (across-wind)

[Figure]

*Figure 1 re-sampling of the hlegs for the July 18, 2017 case. Upper panel: The vertical flight track plotted on radar reflectivity measurements. Lower panel: an example to show that after the resampling, the new hleg #6 and #7 each corresponds one side of the "v" shape flight.*

*Table 1 The selected hlegs after the resampling*

|  | West side of "v" shape (along wind) | East side of "v" shape (across wind) |
|---|---|---|
| New hlegs | 6, 9, 10, 13, 16, 19, 20 | 7, 8, 11, 12, 17, 18, 21 |

After the resampling, the first thing we checked is whether the two modes of the bimodal joint PDF actually correspond to the two sides of the "v" shape flight. And indeed, it is the case.

Then, we repeated the same analysis as we did in Figure 4 and 6 of the original paper for the new hlegs, now with the two sides of the "v" shape flight separated. The key results are summarized in Figure 2. Evidently, both the west side hlegs (Figure 2 a-c) and east side hlegs (Figure 2 d-f)

demonstrate the same vertical structures which are also consistent with those shown in Figure 4 and 6 of the paper: 1) the inverse relative variances $v_q$ and $v_N$ first increases from cloud base to cloud top and then decrees. 2) $q_c$ and $N_c$ become increasingly correlated from cloud base to cloud top. And 3) the total EF decreases from cloud base to cloud top.

These results clearly demonstrate that our conclusions are robust regardless whether or not we separate the "v" shape flight into two sides. We would like to hold onto the original results and plots based on the whole "v" shape flight in the manuscript for two reasons: 1) ~60 km spatial scale is more relevant to the current GCMs and 2) the bimodal joint PDF is possible and its implications for EF and thereby warm rain simulation should be discussed.

[Figure]

Figure 2 a) vertical variations of a) $v_q$ and $v_N$ b) $q_c$ and $N_c$ correlation, and c) the total EF for the new hlegs 6, 9, 10, 13, 16, 19, 20 (west side of the "v" shape flight); d) e) and f) are same as a), b) c), respectively, except or for the new hlegs 7, 8, 11, 12, 17, 18, 21 (east side of the "v" shape flight).

*2) Flying in and out of cloud?*
*How are data for those legs where the aircraft goes in and out of cloud being handled? I assume that you didn't include a lot of zeros in either qc and/or N when calculating the means and variances, or for that matter, the directly calculating E values (e.g., equation 9)? Please describe what was done and comment on the impact this might have on the results.*
**Reply**: As explained in the section 4.1, we use $q_c>0.01$ gm[-3] as the threshold to mask cloudy against clear-sky observations. All of our analyses are based on in-cloud measurements. In other words, we exclude zero values in the computations of $v_q$, $v_N$, $E_q$, $E_N$ or $E$, etc. For example, the $P(q_c)$ in Eq. (6) is the in-cloud PDF of $q_c$ in a GCM grid. We have explicitly point this out at the beginning of Section 4.1. In addition, to avoid potential confusion, we have replaced the lower limit of the integration in Eq. 3, 9 and 10 from zero to minimal in-cloud values $q_{c,min}$ and $N_{c,min}$.

*3) Inner Scale Dependence*
*One of the most interesting points in the paper is that the PDF of N is not lognormally distributed near the boundaries. It seems to me that this is likely due to entrainment*

*preferentially evaporating the smallest cloud droplets first. The time series is very spikey in nature (spikes having low N values). This suggests to me that this effect might be greatly reduced if you started from a larger inner scale (100 or 200 m) rather than the ~10 m.*

*I think it would be a nice addition to your analysis to first average the FCDP data to a scale of 100 to 200 m (use 1 Hz data) and examine how this impacts the skewness of the distributions and the ability of the bivariate log normal to model the variability and enhancement factors.*

**Reply**: Following your suggestion, we did a sensitivity study using 1 Hz FCDP data (instead of the original 10 Hz) for the analysis. As an illustrative example, Figure 3 shows $q_c$ *and* $N_c$ *time series based on 1 Hz (upper panel) vs. 10 Hz (lower panel) FCDP observations* for the hleg #8 at cloud top.

[Figure]

[Figure]

*Figure 3 $q_c$ and $N_c$ time series based on 1 Hz (upper panel) vs. 10 Hz (lower panel) FCDP observations.*

Using the 1 Hz data, we repeated all the analyses we did for the July 18, 2017 case. As shown in Figure 4 and Figure 5, the results are almost identical to those based on 10 Hz data. Therefore, we can conclusions hold for both 10 Hz and 1 Hz FCDP data.

[Figure]

*Figure 4 inverse relative variances based on* a) 1Hz and b) 10 Hz

*Figure 5 Analysis of the En bias for hleg 10 based on 1 Hz FCDP data (a and b) vs. 10 Hz FCDP data (c and d).*

*4) Dependency rather than trend*
*I think the discussion is overly focused on whether there is a (linear) trend in the data. Let's look at Figure 9. To me, "trend" just doesn't seem like the right word to describe the situation in Figure 9a. I suggest a more apt description might be, "Figure 9a shows significant scatter in E with altitude, with smaller values and relatively little scatter near cloud top, and decidedly larger values near cloud base."*

*I understand entirely that the change with height projects on to a line and the correlation is significant. There is a vertical dependence. But it is not obvious to me that there is a linear dependence or trend.*

*Importantly I see Figure 9c has this same pattern, with larger En values (on average) at the bottom than the top, as does 9b for Eq with the exception of the two outliers near cloud top.*

*Somewhat similarly in 9d, the correlations are clearly stronger and there is less variability near cloud top. But it is not clear to me there is a linear dependence. I note the bimodal points are causing much of the variability mid-cloud.*

*I recognize this is partly philosophical, but I think it would be better to recast some of the discussion in terms of a there being a vertical dependence rather than a "trend", though perhaps with more points (general comment 1b) a trend will become clearer.*

**Reply**: Good point! We put all the cases together in Figure 9 in hoping to see if there is any common feature. And the only common features we could observe are the decreasing "trends" in $E$ (Figure 9 a) and increasing "trend" in $\rho$ (Figure 9d). But we agree with your observation that it is not very convincing to call it a trend. Indeed, the vertical variation of $v_q$, $v_N$, and thereby EF are unlikely going to be linear. On the other hand, because of the limited sampling rate and the big differences among the selected cases, it is hard, if not impossible, to resolve both horizonal and at the same time the detailed vertical variations of cloud properties using the in situ measurements. We revised the manuscript following your suggestions and also pointed out the caveats when interpreting the results in Figure6 and 9 (at the end of Section 4.1).

*Specific Comments:*
*Line 213. What is the WCM-2000? Please provide a reference and perhaps add to table #1. Perhaps add a figure demonstrating agreement in qc, or at least quantify what "excellent" agreement means (e.g. Bias less than X and RMS difference less than Y% at 10 Hz?). I note here that this study is about variability and so ideally it would be good to establish variability in qc is the same from both sensor (not simply that the bias between two measurements is small).*

**Reply**: WCM stands for "Water Content Measurement" ("2000" is just a model number). We added a reference (Alyssa and Mei) for this instrument. We compared the qc from FCDP with the WCM measurements for the selected hlegs of the July 18, 2017 case. The mean values of the two measurements are generally within 20% (this information is added to the manuscript as suggested). We have consulted the DOE measurement team (i.e., Fan Mei) and confirmed that the difference is reasonable. The two instruments seem to have a time lag, probably due to the instrument response difference. The instrument differences are beyond the scope of this study.

*Table #1. Perhaps change label "Accuracy" to "Particle Size Intervals" or something similar?*
**Reply**: Good suggestion. We changed to "size resolution".

*Line 214. A value of 20 microns for* ›`IS´ §* *is about as small a threshold as might be chosen, and there are often a significant number of particles in the 20 to 50 micron size range for clouds that are described as "non precipitating". I expect particle concentrations in the 20 to 50 micron size range will co-vary with the concentration of particles less than 20, so I am not surprised that you see little sensitivity. Nonetheless I think it would have been better test of the sensitivity by*

*choosing values for r\* of 20 and 50, rather than 20 +/-5. In particular, you show later in the manuscript that N (with the cut off of r\* of 20) is not well described by a lognormal near cloud-top or cloud-base. Is this result sensitive to the choice of r\*?*

**Reply**: The $r^*$ is the threshold used to separate the "cloud mode" and "precipitation mode", and also the autoconversion and accretion processes. As explained in the paper, we choose $r^*$=20 μm to follow previous studies (e.g., Wood 2005). $r^*$=50 μm seems too large too us. Nevertheless, we did a sensitivity study in which we set $r^*$=50 μm and got almost identical results (see Figure 6 below). Your expectation is correct. The results are not sensitive to the choice of $r^*$.

[Figure]

*Figure 6 Inverse relative variance plot when set $r^*$=50 μm*

*Line 238. Do I understand correctly that only flight labeled as drizzling in table #2 were considered? If so, how were the groupings in Table #2 established? Does "non precipitating" in table #2 mean only particles less than 20 microns where present OR Only when drizzle is clearly falling from clouds (from radar and "pilots" on line 269)?*

*Perhaps it doesn't matter for this paper, but I presume models will apply autoconversion rate parameterizations all the time and I think it would be useful to examine precipitation formation for cloud without obvious precipitation / virga falling from the bottom OR otherwise clearly establish the conditions under which your results apply.*

**Reply**: Reviewer #2 raised somewhat similar questions. At the beginning, we used the pilot summary in table #2 to narrow down our search for heavily drizzling cases. Then, we manually selected the 4 cases mainly based on radar observations (i.e., strong radar reflectivity with precipitation reaching surface). We didn't select the non-precipitating cloud cases for a couple of reasons. The first reason is that we would like to make sure that the drizzling processes, including both autoconversion and accretion have been initialized in the selected case. The second reason is more practical. It is because non-precipitating clouds are usually physically thinner than precipitating clouds, which makes it difficult for the airplane to sample different vertical locations of the clouds. As a result, there is often only one or two in-cloud hlegs for the non-precipitating clouds.

Nevertheless, per your suggestion, we selected three non-precipitating or weakly precipitating cases: 1) 2017-07-13 (non-precipitating) 2) 2018-01-26 (weakly precipitating at cloud base but no perception on the ground) 3) 2018-02-07 (very weakly precipitating at cloud base but no perception on the ground) and added them to the revised manuscript. The radar reflectivity curtain with vertical flight track for these three cases are shown in Figure 7 below. The abovementioned challenge of sampling thin non-precipitating cloud can be clearly seen in Figure 1a for the 2017-07-13 case. The selected hlegs and vlegs for these cases are summarized in Table 2. We repeated the same analyses for these new cases as for other cases, i.e., the vertical and horizontal structures of qc and Nc, as well as the EF, for these newly added cases. Overall, the results from these newly added non-precipitating cloud cases are highly similar to those based on the July 18, 2017 case as discussed in section 4. Take the 2018-02-07 case for example. Figure 8 shows the vertical variation of the inverse relative variances $v_q$ and $v_N$. Apparently, both $v_q$ and $v_N$ demonstrate a pattern similar to that of the July-18, 2017 case (see Figure 4c of the paper), i.e., increasing first from cloud base (hleg 1 -> hleg 2) and then decrease toward cloud top (hleg 3). Therefore, these newly added cases do not affect the general conclusion although they add to the statistics.

[Figure]

*Figure 7 Three non-precipitating (or weakly precipitating) clouds added to the revised manuscript.*

*Table 2 A summary of selected RFs, and the selected hlegs and vlegs within each RF. (newly added non-precipitating cases are highlighted in bold font)*

| Research Flight | Precipitation | Sampling pattern | Selected hlegs | Selected vlegs |
|---|---|---|---|---|

| July 13, 2017 | Non- Precipitating | Straight-line | 3, 4, 5, | 0, 1, 3 |
|---|---|---|---|---|
| July 18, 2017 | Precipitation reaching ground | "V" shape | 5, 6, 7, 8, 10, 11, 12 | 0, 1, 3 |
| Jan. 19, 2018 | Precipitation reaching ground | "V" shape | 6, 7, 8, 15, 16 | 0, 1, 3 |
| July 20, 2017 | Precipitation reaching ground | "V" shape | 5, 6, 7, 8, 9, 13, 14 | 0, 1 |
| **Jan. 26, 2018** | **Precipitation only at cloud base** | **Straight-line** | **3, 4, 5, 9, 10, 11** | **0, 1, 3** |
| **Feb. 07, 2018** | **Non- Precipitating** | **"V" shape** | **1, 2, 3, 5** | **0, 1** |
| Feb. 11, 2018 | Precipitation reaching ground | Straight-line | 4, 5, 6, 7, 12, 13 | 0, 1 |

[Figure]

*Figure 8 vertical dependence of the inverse relative variances for the Feb. 07 2018 case.*

*Line 269. I don't fully understand this criteria or how it is being applied. Is this requirement applied to each horizontal leg or just for choosing cases (see comment for line 238)? If individual legs, are you focusing on the portion of the leg that occurs near the*

*radar? Is there a reflectivity threshold? In general, you have CDP and 2DS data and so I don't understand the choice to rely on radar or pilots.*

**Reply**: Since we added the non-precipitating cloud cases, this criterion *is* removed. For your information, the criterion was originally applied to research flights in table #2 rather than hlegs.

*Line 274. Does "same region" here mean the hlegs must occur along roughly the same track, that is the same set of latitudes and longitudes? If yes, I don't understand the rationale for this. As long as you are sampling the same cloud field (i.e. it is reasonable to expect a representative measure of the variability) and you are using hlegs with the same total length, why do you care if the hlegs and nominally stacked?*

**Reply**: By "*same region*" we mean that the selected hlegs should be within the same "virtual" GCM grid box. The hlegs do not have to be "*nominally stacked*". Take the July 18, 2017 case for example. All the "v"-shape legs naturally formed a "virtual" grid-box. Although the airplane repeated many "v"-shape legs, there are also long transient flights (See Figure 1a of the paper). We excluded these flights because they are outside of the "virtual" grid-box.

*Line 332. Previous studies? Please provide a reference.*

**Reply**: The following references are added: Barker 1996, Lebsock et al. 2013 and Zhang et al. 2019.

*Line 443. "slight"? It is not clear to me whether this difference would or would not have a significant impact on a model simulation. Perhaps just indicate the bivariate lognormal results in an EF value that is about 0.5 larger (be quantitative rather than qualitative).*

**Reply**: Thanks for raising this question. Actually, after examination we found a small bug in our codes (see explanation at the beginning). The Eq based on lognormal parameterization are actually very close to observation-based values, which is consistent with our previous finding in Zhang et al. 2019. The mean bias is only 0.06. this result. This information is added to the revised paper.

*Line 541. It seems natural to expect qc and N will be positively correlated, since stronger updraft mean both activating more droplets and condensing more water.*

**Reply**: Yes, this is one possible reason. Another possibility is the inhomogeneous mixing due to cloud top entrainment, which reduces the qc and Nc simultaneously. In this study, we can only speculate about the possible reasons. At the moment, we are also using LES to investigate the underlying physics.

*Line 503-12. The use of a "relative hleg altitude" seem to me to be "just as prone" or even "more prone" to misinterpretation as using cloud boundaries from either radar/lidar or the vlegs. Here you are effectively demanding that the highest and lowest legs in each flight should have the same relationship regardless of how close or far they are from the actual cloud top or cloud base. I guess the question is, does it matter? Based on your 4th criteria (line 281) I would hope not.*

**Reply**: As one can see from Figure 1 of the paper, the selected cases are quite different in terms of cloud thickness and cloud boundaries. So, to make any meaningful comparison, we have to first normalize the boundary of each case, which lead us to the use of *"relative hleg altitude"*.

We can the cloud boundaries either from in-situ or radar/lidar measurements for the normalization, although uncertainties are inevitable either way.

Due the field campaign, the pilot was actually instructed to sample the cloud base and cloud top as close as possible because these observations are very useful for studying clouds (e.g., understanding cloud top entrainment and cloud base precipitation). However, it is still extremely difficult to determine cloud boundaries from in situ measurements alone. This is why we have required that the in situ and radar/lidar measurements are largely consistent (i.e., the *4th criteria (line 281)")*

*Minor Comments*
*Line 55-6. Replace ". . . at the spatial scales much smaller .. " with ". . . on spatial scales that are much smaller . . ."*
**Reply**: Done

*Line 56-7. Remove the phrase " . . . making the simulation of these processes in GCMs highly challenging." This is entirely redundant with the "challenging" remark on the previous sentence.*
**Reply**: Done

*Line 164. Perhaps rephrase as ". . . better understand the horizontal variations in*
**Reply**: ok done.

*Line 184. Replace "seasonable" with "seasonal"*
**Reply**: Done.

*Line 185. Perhaps add comma and remove "and" so the sentence reads " . . . the MBL (Dong et al., 2014; Rémillard 185 et al., 2012), to improving cloud parameterizations in the GCMs (Zheng et al., 2016), to validating the space-borne remote sensing products of MBL clouds (Zhang et al., 2017)."*
**Reply**: Good suggestion. Done.

*Line 189. Change "is" to "was", and perhaps simplify to read "In 2013 a permanent measurement site was established by the ARM program on Graciosa Island, and is typically referred to as the ENA site (Voyles and Mather, 2013)."*
**Reply**: Done.

*Figure 1. I presume the green blobs are islands? Perhaps explain this in the caption along with note about location of the ARM site and likewise describe what the number in white boxes and colored stripes mean in radar panels? Also I suggest higher resolution figure would be helpful here.*

**Reply**: Yes, they are islands. Figure 1 has been revised following your suggestions and figure captions updated. Note that we have moved the figures for other selected cases to the supplementary materials so we can use high-resolution and larger figures for July 18, 2017 case.

*Figure 9. The solid symbols seem a bit problematic here. I am pretty sure that in panel c, there is a red dot near cloud top that must be hidden under the other symbols – and this make me wonder if other symbols might also be hidden.*

**Reply**: we revised the figure using open symbols to reduce overlap.

*Line 537. I think you mean "Eq is larger than E", and you need to change the second occurrence of Eq in this sentence to be simply E.*
**Reply**: yes, you are right. We corrected this.

*Line 561. Perhaps change to read "... implications of subgrid variability as relates to the enhancement of autoconversion rates ..."*
**Reply**: Thanks for the suggestion. We revised accordingly.

*Line 566. Perhaps add "near cloud top" to the end of the sentence. I think it is reasonable to make the association with cloud-top entrainment but obviously, the study doesn't define entrainment zone as best I recall.*
**Reply**: *Changed to "near cloud top".*

*Line 569. Change "we" to "our".*
**Reply**: Done.

**Vertical Dependence of Horizontal Variation of Cloud Microphysics:**
**Observations from the ACE-ENA field campaign and implications for warm rain**
**simulation in climate models**

Zhibo Zhang[1,2,*], Qianqian Song[1,2], David B. Mechem[3], Vincent E. Larson[4], Jian Wang[5],

Yangang Liu[6], Mikael K. Witte[7,8], Xiquan Dong[9], Peng Wu[9,10]

1.  Physics Department, University of Maryland Baltimore County (UMBC), Baltimore, 21250, USA

2.  Joint Center for Earth Systems Technology, UMBC, Baltimore, 21250, USA

3.  Department of Geography and Atmospheric Science, University of Kansas, Lawrence, 66045, USA

4.  Department of Mathematical Sciences, University of Wisconsin — Milwaukee, Milwaukee, 53201, USA

5.  Center for Aerosol Science and Engineering, Department of Energy, Environmental and Chemical Engineering, Washington University in St. Louis, St. Louis, 63130, USA

6.  Environmental and Climate Science Department, Brookhaven National Laboratory, Upton, 11973, USA

7.  Joint Institute for Regional Earth System Science and Engineering, University of
California Los Angeles, Los Angeles, 90095, USA

8.  Jet Propulsion Laboratory, California Institute of Technology, Pasadena, 91011, USA

9.  Department of Hydrology and Atmospheric Sciences, University of Arizona, Tucson, 85721, USA

10. Pacific Northwest National Laboratory, Richland, WA 99354, USA

To be submitted to the ACP special issue:  Marine aerosols, trace gases, and clouds over the North Atlantic

*Correspondence to*: Zhibo Zhang (zhibo.zhang@umbc.edu)

**Abstract:**

In the current global climate models (GCM), the nonlinearity effect of subgrid cloud variations on the parameterization of warm rain process, e.g., the autoconversion rate, is often treated by multiplying the resolved-scale warm ran process rates by a so-called enhancement factor (EF). In this study, we investigate the subgrid-scale horizontal variations and covariation of cloud water content ($q_c$) and cloud droplet number concentration ($N_c$) in marine boundary layer (MBL) clouds based on the in-situ measurements from a recent field campaign and study the implications for the autoconversion rate EF in GCMs. Based on a few carefully selected cases from the field campaign, we found that in contrast to the enhancing effect of $q_c$ and $N_c$ variations that tends to make EF>1, the strong positive correlation between $q_c$ and $N_c$ results in a suppressing effect that tends to make EF<1. This effect is especially strong at cloud top where the $q_c$ and $N_c$ correlation can be as high as 0.95. We also found that the physically complete EF that accounts for the covariation of $q_c$ and $N_c$ is significantly smaller that its counterpart that accounts only for the subgrid variation of $q_c$, especially at cloud top. Although this study is based on limited cases, it suggests that the subgrid variations of $N_c$ and its correlation with $q_c$ both need to be considered for an accurate simulation of the autoconversion process in GCMs.

[revised manuscript text omitted]
, which is used in this study. We have also done a sensitivity study, in which we averaged the FCDP data to 1 Hz and got almost identical results. Since the typical horizontal speed of the G-1 aircraft during the in-cloud leg is about 100 m s$^{-1}$, the spatial sampling rate these instruments is on the order of 10 m for the FCDP at 10 Hz.

**2.2. Ground observations from ARM ENA site**

In addition to the in-situ measurements, ground measurements from the ARM ENA site are also used to provide ancillary data for our studies. In particular, we will use the Active Remote Sensing of Cloud Layers product (ARSCL; *(Clothiaux et al., 2000; Kollias et al., 2005)* which blends radar observations from the Ka–band ARM zenith cloud radar (KAZR), micropulse lidar (MPL), and the ceilometer to provide information on cloud boundaries and the mesoscale structure of cloud and precipitation. The ARSCL product is used to specify the vertical location of the G1 aircraft and thereby the in-situ measurements with respect to the cloud boundaries, i.e., cloud base and top (see example in Figure 1). In addition, the radar reflectivity observations from KAZR, alone with in situ measurements, are used to select the precipitating cases for our study. Note that
* * *
the ARSCL product is from the vertically pointing instruments, which sometimes are not collocated with the in-situ measurements from G1 aircraft. As explained later in the next section, only those cases with a reasonable collocation are selected for our study.

**3. Case selections**

**3.1. ACE-ENA flight pattern**

The section provides a brief overview of the G1 aircraft flight patterns during the ACE-

ENA and explains the method for cases selections for our study using the July 18, 2017 RF as an example. As shown in Table 2, a variety of MBL conditions were sampled during the two IOPs of the ACE-ENA campaign, from mostly clear-sky to thin stratus and drizzling stratocumulus. The basic flight patterns of G1 aircraft in the ACE-ENA included spirals to obtain vertical profiles of aerosol and clouds, and legs at multiple altitudes, including below cloud, inside cloud, at the cloud top, and in the free troposphere. As an example, Figure 1a shows the horizontal location of the G1

aircraft during the July 18, 2017 RF which is the "golden case" for our study as explained in the next section. The corresponding vertical track of the aircraft is shown in Figure 1b overlaid on the reflectivity curtain of ground based KAZR. In this RF, the G1 aircraft repeated multiple times of horizontal level runs in a "V" shape at different vertical levels inside, above and below the MBL

(see Figure 1b). The lower tip of the "V" shape is located at the ENA site on Graciosa island. The average wind in the upper MBL (i.e., 900 mb) is approximately Northwest. So, the west side of the V-shape horizontal level runs is along the wind and the east side across the wind. Note that the horizontal velocity of the G1 aircraft is approximately 100 m s$^{-1}$. Since the duration of these selected "V" shape hlegs is between 580 s and 700 s, their total horizontal length is roughly 60 km, with each side of the "V" shape ~30 km. These "V" shape horizontal level runs, with one side along and the other across the wind, are a common sampling strategy used in the ACE-ENA to

Deleted: In this study, we are interested in the RFs that encountered the drizzling stratocumulus clouds, since our objective is to understand the implications of subgrid cloud variation for the autoconversion process.

observe the properties of aerosol and cloud at different vertical levels of the MBL. In our study we use the vertical location of the G1 aircraft from the AIMMS to identify continuous horizontal flight tracks which are referred to as the "hleg". For the July 18, 2017 case, a total of 13 hlegs are identified as shown in Figure 1b. Among them, the hleg 5, 6, 7, 8, 10, 11, and 12 are the seven V- shape horizontal level runs inside the MBL cloud. Together they provide an excellent set of samples of the MBL cloud properties at different vertical levels of a "virtual" GCM grid box of about 30 km. As aforementioned, Boutle et al. (2014) found that the horizontal variance of $q_c$

increases with the horizontal scale $L$ slowly when $L$ is larger than about 20 km. Therefore, although the horizontal sampling of the selected hlegs is only about 30 km, the lessons learned here could yield useful insights for larger GCM grid sizes. In addition to the hlegs, we also identified the vertical penetration legs in each flight, referred to as the "vlegs", from which we will obtain the vertical structure of the MBL, along with the properties of cloud and aerosol.

**3.2. Case selection**

As illustrated in Figure 1 a and b for the July 18, 2017 RF, the criteria we used to select the

RF cases and the hlegs within the RF can be summarized as follows:

• The RF samples multiple continuous in-cloud hlegs at different vertical levels with the horizontal length of at least 10 km and cloud fraction larger than 10% (i.e., the fraction of a hleg with $q_c>0.01$g m$^{-3}$ must exceed 10% of the total length of that hleg). It is important to note here that, unless otherwise specified, all the analyses of $q_c$ and $N_c$ are based on in- cloud observations (i.e., in the regions with $q_c>0.01$g m$^{-3}$).

• Moreover, the selected hlegs must sample the same region (i.e., the same virtual GCM grid box) repeatedly in terms of horizontal track but different vertical levels in terms of vertical track. Take the July 18, 2017 case as an example. The hleg 5, 6, 7, 8 follow the same "V"

shape horizontal track (see Figure 1a) but sample different vertical levels of the MBL

clouds (see Figure 1b). Such hlegs provide us the horizontal sampling needed to study the subgrid horizontal variations of the cloud properties and, at the same time, the chance to study the vertical dependence of the horizontal cloud variations.

• Finally, the RF needs to have at least one vleg and the cloud boundary derived from the vleg is largely consistent with that derived from the ground-based measurements. This requirement is to ensure that the vertical locations of the selected hlegs with respect to cloud boundaries can be specified. For example, as shown in Figure 1b according to the ground-based observations, the hlegs 5 and 10 of July 18, 2017 case are close to cloud base, while hlegs 8 and 12 close to cloud top (see also Figure 4).

The above requirements together pose a strong constraint on the observation. Fortunately, thanks to the careful planning of the RF which had already taken studies like ours into consideration, we are able to select a total of seven RF cases as summarized in Table 3. The plots of the flight tracks and ground-based radar observations for the six other RF cases are provided in the supplementary material (Figure S1—S6). 
[revised manuscript text omitted]
. It is unlikely that the bimodality is caused by the along-wind and across-wind difference between the two sides of the "V" shape track. It is most likely just a coincidence. On the other hand, the bimodal joint distribution between $q_c$ and $N_c$ is "real" which could be a result of subgrid variations of updraft, precipitation and/or aerosols.

As a result of the bimodality of $N_c$, the correlation coefficients between $q_c$ and $N_c$ is significantly smaller for the hlegs 6 ($\rho = 0.22$) and 7 ($\rho = 0.31$) in comparison with other hlegs.

However, if the two sides of the "V" shape tracks are considered separately, then the $q_c$ and $N_c$

become more correlated, except for the east side of hleg 6 which still exhibits to some degree a bimodal joint distribution of $q_c$ and $N_c$. In spite of the bimodality, there is evidently a general increasing trend of the correlation between $q_c$ and $N_c$ from cloud base toward cloud top. At the cloud top, the $q_c$ and $N_c$ correlation coefficient can be as high as $\rho = 0.95$ for hleg 12 (see Figure

5e). As explained in the next section, this close correlation between $q_c$ and $N_c$ has important implications for the simulation of autoconversion enhancement factor.

As a summary, the above phenomenological analysis of the July 18, 2017 RF reveals the following features of the horizontal and vertical variations of cloud microphysics. Vertically, the mean values of $q_c$ and $N_c$ qualitatively follow the adiabatic structure of MBL cloud, i.e., $q_c$

increases linear with height and $N_c$ remains largely invariant above cloud base. Even though the joint distribution of $q_c$ and $N_c$ exhibits a bimodality in several hlegs, their correlation generally increases with height and can be as high as $\rho = 0.95$ at cloud top. Horizontally, both $q_c$ and $N_c$

have a significant variability at cloud base, which tends to first decrease upward and then increase in the uppermost part of cloud close to the entrainment zone. Finally, we have to point out a couple of important caveats in the above analysis. First, as seen from Figure 1 the selected hlegs are sampled at different vertical locations and also at different time. For example, the hleg 5 at cloud base is more than 1 hour apart from the hleg 8 at cloud top (Figure 1a). As a result, the temporal evolution of clouds is a confounding factor and might be misinterpreted as vertical variations of clouds. On the other hand, as shown below, we also observed similar vertical structure of $q_c$ and

$N_c$ in other cases. It seems highly unlikely that the temporal evaluations of the clouds in all selected cases conspire to confound our results in the same way. Based on this consideration, we assume that the temporal evolution of clouds is an uncertainty that could lead to random errors but does not impact the overall vertical trend. The second caveat is that due to the very limited vertical sampling rate of hlegs (i.e., only 3-4 samples) we cannot possibly resolve the detailed vertical variation of $\nu_q$, $\nu_N$ and $\rho$. Although we have used the word "trend" in the above analysis, it should be noted that the vertical profile of these parameters may, but more likely may not, be linear. So, the word "trend" here indicates only the large pattern that can be resolved by the hlegs. Obviously these two caveats also apply to the analysis below of the EF which is also derived from the hlegs.

**4.2. Implications for the EF for the autoconversion rate parameterization**

As explained in the introduction, in GCMs the autoconversion process is usually parameterized as a highly nonlinear function of $q_c$ and $N_c$, e.g., the KK scheme in Eq. (1). In such parameterization, an EF is needed to account for the bias caused by the nonlinearity effect. A

variety of methods have been proposed and used in the previous studies to estimate the EF *(Larson*

*and Griffin, 2013; Lebsock et al., 2013; Pincus and Klein, 2000; Zhang et al., 2019)*. The methods used in this study are based on Z19. Only the most relevant aspects are recapped here. Readers are referred to Z19 for detail.

If the subgrid variations of $q_c$ and $N_c$, as well as their covariation, are known, then the EF

can be estimated based on its definition as follows

$$E = \frac{\int_{N_{c,min}}^{\infty} \int_{q_{c,min}}^{\infty} q_c^{\beta_q} N_c^{\beta_N} P(q_c, N_c) dq_c dN_c}{\langle q_c \rangle^{\beta_q} \langle N_c \rangle^{\beta_N}}, \qquad (3)$$

where $\langle q_c \rangle$ and $\langle N_c \rangle$ are the grid-mean value, $P(q_c, N_c)$ is the joint probability density function (PDF) of $q_c$ and $N_c$. $q_{c,min}$ and $N_{c,min}$ are the lower limits of the in-cloud value (e.g., $q_{c,min}$=0.01

[revised manuscript text omitted]

uncertainty and a confounding factor in this study, which needs to be quantified in future studies.

Third, our study provides only a phenomenological analysis of the horizontal variations cloud microphysics in the MBL clouds and the implications for the EF. Ongoing modeling research based on a comprehensive LES model is being conducted to identify and elucidate the process- level physical mechanisms behind our observational results. Finally, this study is focused on the

KK parameterization in estimating the enhancement factors resulting from subgrid variability of

$q_c$, $N_c$ and $q_c$-$N_c$ covariance. The specific values are expected to differ when applied to other autoconversion parameterizations with different power-law exponents.

**Acknowledgement:**

Z. Zhang acknowledges the financial support from the Atmospheric System Research (Grant DE-

SC0020057) funded by the Office of Biological and Environmental Research in the US DOE Office of

Science. The computations in this study were performed at the UMBC High Performance Computing

Facility (HPCF). The facility is supported by the U.S. National Science Foundation through the MRI

program (Grants CNS-0821258 and CNS-1228778) and the SCREMS program (Grant DMS-0821311), with substantial support from UMBC. Co-author D. Mechem was supported by subcontract OFED0010-

01 from the University of Maryland Baltimore County and the U.S. Department of Energy's Atmospheric

Systems Research grant DE-SC0016522.

*Table 1 In situ cloud instruments from ACE-ENA campaign used in this study*

| Instruments | Measurements | Frequency | Resolution | Size resolution |
|---|---|---|---|---|
| **AIMMS** | P, T, RH, u,v,w | 20 Hz | / | / |
| **F-CDP** | DSD 2~50 µm | 10 Hz | 1 -2 µm | 2 µm |
| **2DS** | DSD 10 ~2500 µm | 1 Hz | 25 – 150 µm | 10 µm |

Table 2  *conditions of MBL sampled during the two IOPs of ACE-ENA campaign*

| Conditions Sampled | Research Flights | |
|---|---|---|
| | IOP1: June-July 2017 | IOP2: Jan.-Feb. 2018 |
| Mostly clear | 6/23, 6/29, 7/7 | 2/16 |
| Thin Stratus | 6/21, 6/25, 6/26, 6/28, 6/30, 7/4, 7/13 | 1/28, 2/1, 2/10, 2/12 |
| Solid StCu | 7/6, 7/8, 7/15 | 1/30, 2/7 |
| Multi-layer StCu | 7/11, 7/12 | 1/24, 1/29, 2/8 |
| Drizzling StCu/Cu | 7/3, 7/17, 7/18, 7/19, 7/20 | 1/19, 1/21, 1/25, 1/26, 2/9, 2/11, 2/15, 2/18, 2/19 |

Table 3 A summary of selected RFs, and the selected hlegs and vlegs within each RF.

| Research Flight | Precipitation | Sampling pattern | Selected hlegs | Selected vlegs |
|---|---|---|---|---|
| July 13, 2017 | Non- Precipitating | Straight-line | 3, 4, 5 | 0, 1, 3 |
| July 18, 2017 | Precipitation reaching ground | "V" shape | 5, 6, 7, 8, 10, 11, 12 | 0, 1, 3 |
| July 20, 2017 | Precipitation reaching ground | "V" shape | 5, 6, 7, 8, 9, 13, 14 | 0, 1 |
| Jan. 19, 2018 | Precipitation reaching ground | "V" shape | 6, 7, 8, 15, 16 | 0, 1, 3 |
| Jan. 26, 2018 | Precipitation only at cloud base | Straight-line | 3, 4, 5, 9, 10, 11 | 0, 1, 3 |
| Feb. 07, 2018 | Non- Precipitating | "V" shape | 1, 2, 3, 5 | 0, 1 |

| Feb. 11, 2018 | Precipitation reaching ground | Straight-line | 4, 5, 6, 7, 12, 13 | 0, 1 |
|---|---|---|---|---|

Formatted Table

Inserted Cells

[Figure]

[Figure]

Figure 1 **(a)** horizontal flight track of the G1 aircraft (red) during the July 18, 2017 RF around the DOE ENA site (yellow star) on the Graciosa Island. The **(b)** vertical flight track of G1(thick black line) overlaid on the radar reflectivity contour by the ground-based KZAR. The dotted lines in the figure indicate the cloud base and top retrievals from ground-based radar and CEIL instruments. The yellow shaded regions are the "hlegs" and green shaded regions are "vlegs". See text for their definitions.

[Figure]

a) MODIS Satellite Image    b) Sea Level Pressure

[revised manuscript text omitted]